# Induction of hepatitis B core protein aggregation targeting an unconventional binding site

Vladimir Khayenko[1,2†], Cihan Makbul[1,2†], Clemens Schulte[1,2], Naomi Hemmelmann[1,2], Sonja Kachler[1,2], Bettina Böttcher[1,2]*, Hans Michael Maric[1,2]*

[1]Rudolf Virchow Center, Center for Integrative and Translational Bioimaging; University of Würzburg, Würzburg, Germany; [2]Biocenter, University of Würzburg, Würzburg, Germany

**\*For correspondence:**
bettina.boettcher@uni-wuerzburg.de (BB);
Hans.Maric@virchow.uni-wuerzburg.de (HMM)

[†]These authors contributed equally to this work

**Competing interest:** The authors declare that no competing interests exist.

## eLife Assessment

This **valuable** work presents an interesting strategy to interfere with the HBV infectious cycle as it identifies two previously unexplored HBc-Ag binding pockets. The experimental data is **compelling** and opens the door to generating and testing novel anti-HBV therapies.

**Abstract** The hepatitis B virus (HBV) infection is a major global health problem, with chronic infection leading to liver complications and high death toll. Current treatments, such as nucleos(t) ide analogs and interferon-α, effectively suppress viral replication but rarely cure the infection. To address this, new antivirals targeting different components of the HBV molecular machinery are being developed. Here we investigated the hepatitis B core protein (HBc) that forms the viral capsids and plays a vital role in the HBV life cycle. We explored two distinct binding pockets on the HBV capsid: the central hydrophobic pocket of HBc-dimers and the pocket at the tips of capsid spikes. We synthesized a geranyl dimer that binds to the central pocket with micromolar affinity, and dimeric peptides that bind the spike-tip pocket with sub-micromolar affinity. Cryo-electron microscopy further confirmed the binding of peptide dimers to the capsid spike tips and their capsid-aggregating properties. Finally, we show that the peptide dimers induce HBc aggregation in vitro and in living cells. Our findings highlight two tractable sites within the HBV capsid and provide an alternative strategy to affect HBV capsids.

## Introduction

The hepatitis B virus (HBV) infects the liver and can cause acute and chronic hepatitis. In childhood and infancy, the virus is particularly dangerous as the recovery rate among children is approximately 50%, while among infants infected through perinatal transmission, only 10% will naturally recover, the remainder will develop chronic infection (*Thomas, 2019*; *Block et al., 2021*). On the global scale, the most effective approach to address hepatitis B is through preventive treatment with vaccinations. However, the goals of achieving sufficient vaccination coverage and timely immunization have yet to be met (*Thomas, 2019*; *Cox et al., 2020*). Furthermore, vaccinations are ineffective for individuals who are already infected (*Dienstag et al., 1982*). With about 300 million chronic carriers and over 800,000 hepatitis-related yearly deaths, chronic hepatitis B is a global health problem (*Block et al., 2021*; *World Health Organisation, 2022*) that requires a solution.

**eLife digest** New and better strategies to treat hepatitis B are urgently needed. Many people worldwide remain unvaccinated against the disease, leaving them vulnerable to infection and serious liver problems. Children are particularly at risk of developing long-term illness. Treatments exist to help manage the condition, but they can rarely cure it.

Hepatitis B virus is protected by a spiky shell called the capsid, made of HBc proteins. This structure is critical for survival, and therefore a promising therapeutic target. Current approaches rely on compounds disrupting the assembly or stability of this structure by binding onto the HBc protein. So far, most of these drug candidates target the same location at the base of the capsid's spikes. Until now, other binding pockets on the capsid remained largely unexplored.

To investigate whether these sites could be potential drug targets, Khayenko et al. developed two types of molecules, peptides and geranyl dimers, that could theoretically attach to the capsid – the former at the tip of the spikes and the latter in their middle section. In vitro and in living cells, the compounds not only latched onto these sites, but caused HBc proteins to clump together, preventing the capsid from forming properly. A similar mode of action is observed with existing drug candidates that, however, all bind to the pocket at the base of the spikes.

These findings highlight alternative strategies for targeting hepatitis B. Future studies will need to determine how well these molecules work in clinical conditions, and whether they could complement or improve existing treatments.

Currently, there are two approved classes of medications for the treatment of chronic hepatitis B: nucleos(t)ide analogs (NAs) and interferon-α and its derivatives (IFN-α) (*Hepatitis B Foundation, 2023*; *Jeng et al., 2023*). NAs compete for binding with the natural nucleotide substrates, inhibiting the viral protein P in charge of the reverse transcription of the viral pre-genomic RNA (pgRNA) into HBV DNA (*Menéndez-Arias et al., 2014*) IFN-α serves as both an immunomodulator and immunostimulant, activating genes with diverse antiviral functions to target various steps of viral replication. Additionally, it indirectly suppresses HBV infection by modifying cell-mediated immunity (*Liang et al., 2015*).

Present treatments effectively suppress HBV replication, reduce liver inflammation, fibrosis, and the risk of cirrhosis and hepatocellular carcinoma (HCC), but IFN-α treatment is associated with significant adverse effects and NAs typically require long-term oral administration, often lifelong, as treatment discontinuation often results in viral rebound and disease recurrence in many patients. While current therapies manage the disease, a clinical cure is seldom achieved, and the risk of HCC, although reduced, remains (*Jeng et al., 2023*). Consequently, various classes of direct-acting antivirals and immunomodulatory therapies are currently under development, aiming to achieve a functional cure following a finite treatment duration (*Cornberg et al., 2020*).

New HBV antivirals capitalize on the enhanced understanding of the viral life cycle and can be categorized into several classes (*Table 1*): entry inhibitors that disrupt HBV entry into hepatocytes by blocking the binding to the sodium/taurocholate co-transporting polypeptide (NTCP) receptor (*Yan et al., 2012*). HBsAg inhibitors based on nucleic acid polymers that interfere with the production of HBV surface antigens, and viral gene repressors based on nucleases. Translation inhibitors based on small interfering RNAs or antisense oligonucleotides that silence HBV RNA, thereby decreasing the viral antigen production. Finally, the capsid assembly modulators (CAMs) target the hepatitis B core protein (HBc) that participates in multiple essential steps of the HBV life cycle (*Jeng et al., 2023*).

Among the direct acting antivirals that are in preclinical or clinical studies, a third are CAMs (*Hepatitis B Foundation, 2023*). Capsids are attractive targets due to the absence of human homologues for HBc and their involvement in crucial stages of the HBV life cycle, including nuclear entry, encapsulation of the pgRNA and polymerase, optional nuclear recycling to replenish the covalently closed circular DNA (cccDNA) pool, and eventual coating and secretion from infected cells (*Jeng et al., 2023*; *Kim et al., 2021*).

The capsid is composed of 120 units of HBc dimers, assembling into a T = 4 icosahedron. Within this structure, 60 asymmetric units are formed by four HBc monomers each, designated as A, B, C, and D, or AB dimers and CD dimers (*Crowther et al., 1994*). The ultrastructure formed by the HBc dimer

**Table 1.** Direct-acting hepatitis B virus (HBV) antivirals.

| Class | Mechanism of action | Examples | Development stage (Hepatitis B Foundation, 2023) | Molecule type |
|---|---|---|---|---|
| Entry inhibitors | NTCP (sodium/taurocholate co-transporting polypeptide) receptor (*Watashi and Wakita, 2015*) inhibition. | Bulevirtide (*Ligat et al., 2021*) | Phase III | Lipopeptide |
| | | A2342 (*Bonn et al., 2022*) | Preclinical | Small molecule |
| HBsAg inhibitor | Inhibition of the host HSP40 chaperone DNAJB12 that mediates spherical HBV assembly. Reduces the HBsAg in the circulation and lowers intracellular HBsAg (*Vaillant, 2022*). | REP 2139 (*Vaillant, 2019*) | Phase II | Nucleic acid polymer |
| Translation inhibitors | Antisense oligonucleotide (ASO) or small interfering RNAs (siRNA) (*Gareri et al., 2022*) that target HBV messenger RNAs and act to decrease levels of viral proteins. | Bepirovirsen (*Yuen et al., 2022*) | Phase III | ASO |
| | | VIR-2218 (*Gane et al., 2023*) | Phase II | siRNA |
| Viral gene repressors | Specific cleavage mediation of viral covalently closed circular DNA (cccDNA) via nucleases (*Ono and Bassit, 2021*). | PBGENE-HBV (*Gorsuch et al., 2022*) | Preclinical | Endonuclease I-CreI |
| | | EBT107 (*Ono and Bassit, 2021*) | Preclinical | CRISPR-Cas9 |
| Capsid assembly modulators | Target the hydrophobic pocket located at the dimer-dimer interface near the C termini of HBc subunits and induce misassembly of the core protein, thereby impeding the formation of infectious progeny virions (*Ono and Bassit, 2021*; *Kim et al., 2021*; *Zheng et al., 2023*). | Canocapavir (*Zheng et al., 2023*) | Phase II | Small molecule |
| | | EDP-514 (*Feld et al., 2022*) | Phase I | Small molecule |

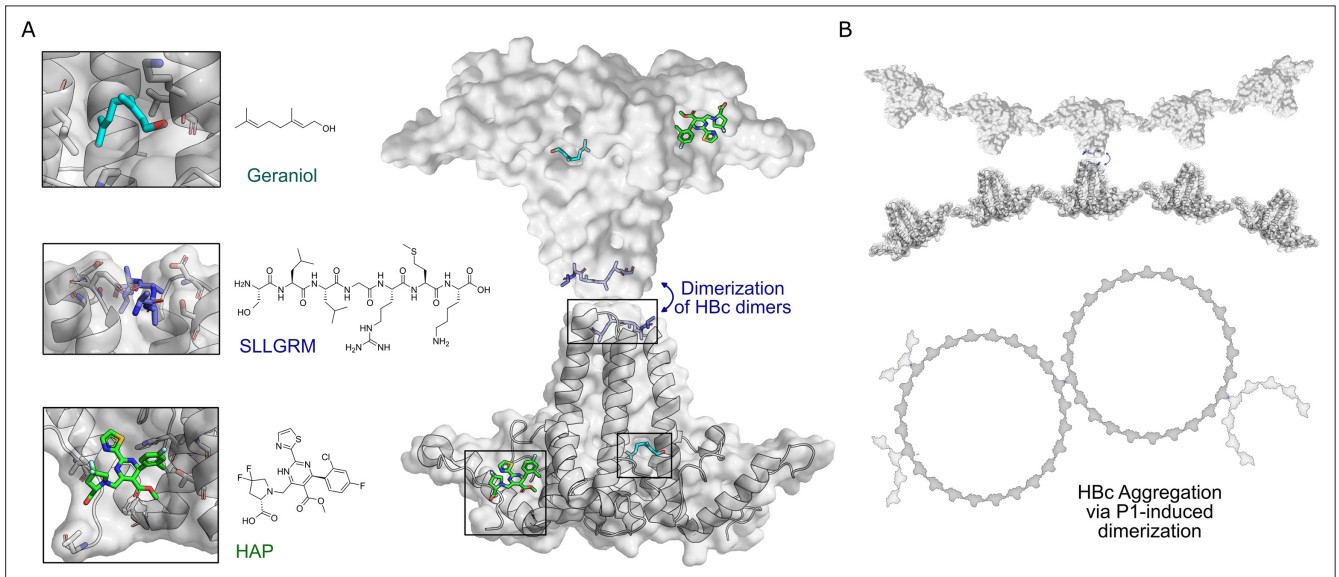

**Figure 1.** HBc binding pockets and mode of action of dimeric binders. (**A**) Left: close-up view of the three addressable effector sites within HBc-dimers (shown as cartoon model with transparent surface in gray) together with representative ligands shown as stick models: SLLGRM peptide (marine blue, PDB: 7PZN); geraniol, resolved here (cyan); heteroaryldihydropyrimidine (HAP [(2S)-1-[[(4R)-4-(2-chloranyl-4-fluoranyl-phenyl)-5-methoxycarbonyl-2-(1,3-thiazol-2-yl)-1,4-dihydropyrimidin-6-yl]methyl]-4,4-bis(fluoranyl)-pyrrolidine-2-carboxylic acid], green, PDB: 5WRE). HAP is a representative example of a canonical capsid assembly modulator (CAM) that targets a hydrophobic pocket mediating HBc-dimer multimerization, an essential step in capsid assembly. A blue arrow indicates how dimeric peptide-based ligands may induce aggregation. Right: the general architecture of an HBc-dimer is depicted as a cartoon with transparent surface model in gray and the three ligands that target distinct binding pockets are in color. The binding sites of two HBc dimers can be linked by dimeric ligands, here exemplified with the peptide ligand. (**B**) Hypothetical mode of action of HBc aggregation triggered by cross-linking the spikes of individual HBc dimers, HBc multimers or the whole capsid.

reveals several binding pockets that can be exploited as potential targets for modulating protein activity (*Figure 1*).

CAMs target the hydrophobic pocket at the HBc dimer–dimer interface, upon binding they strengthen the association energy between HBc-dimer subunits, thereby promoting capsid assembly, rather than inhibiting it (*Venkatakrishnan et al., 2016*; *Stray et al., 2005*). As a result, abnormal or empty capsids may form, sometimes accompanied by the aggregation of core proteins, consequently inhibiting HBV DNA replication. Additionally, CAMs can disrupt the disassembly of incoming virions and the intracellular recycling of capsids, thereby impeding the establishment and replenishment of cccDNA (*Jeng et al., 2023*; *Kim et al., 2021*; *Zoulim et al., 2022*; *Schlicksup and Zlotnick, 2020*).

Recently, a new potentially druggable site was discovered in the HBc dimer—a hydrophobic pocket formed at the base of the spike. This site was targeted by the detergent Triton X-100 (TX100), ultimately causing conformational alterations in the capsid structure (*Lecoq et al., 2021*; *Makbul et al., 2021b*). In addition to the spike base hydrophobic pocket, there is another less-explored interacting domain located on the spike tip of the HBc dimer. Previous studies have shown that peptides targeting the cleft on the spike tip reduced viral replication in a cell model, likely by interfering with viral assembly through modulation of the HBc interaction with the surface antigen (*Böttcher et al., 1998*). These two effector sites could serve as a foundation for the development of new types of HBc modulators and provide alternatives ways for controlling HBV infections.

In this study, we characterize and explore these alternative HBc binding pockets at the inner-dimer interface in the center and at the tips of capsid spikes (*Lecoq et al., 2021*; *Makbul et al., 2021b*; *Makbul et al., 2021a*), and unveil the HBc aggregating properties of a spike-binding dimeric peptide.

## Results

HBV capsid assembly modulation via the binding pockets on the HBc multimer ultrastructure represents a promising pharmacological strategy but until now only one site located on the HBc dimer–dimer interface was explored (*Figure 1A*; *Kim et al., 2021*; *Zheng et al., 2023*). We designed and synthesized bivalent binders specifically targeting two distinct regions of the HBV core protein (HBc)—the hydrophobic pocket at the dimer interface and the tips of the capsid spikes (*Figure 1*). These binders, designed to engage both sites with avidity enhanced affinity, were evaluated for their binding affinity and their effects on HBV capsids in both in vitro assays and living cell models.

### Geranyl dimer targets the central hydrophobic pocket of HBc-dimers with micromolar affinity

Hydrophobic post-translational modifications, such as myristylation of the large hepatitis B virus surface protein (L-HBs), are essential for HBV infectivity and play a role in mediating viral assembly (*Gripon et al., 1995*). Additionally, farnesylation, another hydrophobic post-translational modification, is involved in the envelopment of hepatitis D virus (HDV) (*Koh et al., 2015*), which relies on the presence of HBV and its protein machinery for propagation. Recently, TX100, a nonionic surfactant sharing a similar hydrocarbon binding motif as the natural HBV post-translational modifications, was identified as a ligand of a distinct hydrophobic pocket in the center of HBc-dimers (*Lecoq et al., 2021*; *Makbul et al., 2021b*; *Roseman et al., 2005*).

Several of the pocket forming amino acids, such as K96 and 129-PPAY-132 (*Rost et al., 2006*) and the natural occurring point mutations HBcP5T, L60V, F97L, and P130T (; *Le Pogam et al., 2000*; *Yuan and Shih, 2000*; *Yuan et al., 1999a*; *Ehata et al., 1992*) are involved in the secretion of enveloped virions from the cell. These findings lead to the infectious HBV particles signal hypothesis where this hydrophobic pocket is involved in the regulation of the envelopment of nucleocapsids and thus could be an alternative druggable pocket to block virus envelopment (*Roseman et al., 2005*).

We reasoned that compounds mimicking the natural HBV/HDV compounds and sharing a hydrophobic motif similar to that of TX100 can prove to be potent binders of this key hydrophobic pocket. Therefore, we set out to test *n*-decyl-beta-D-maltopyranoside (DM) (**1**), geraniol (**2**), and its synthesized dimer (**3**), as the mimetics of myristic acid and farnesyl, respectively.

The isothermal calorimetric titration (ITC) of HBc capsids with DM (**1**) resolved micromolar affinity ($K_D$ = 133 ± 38 µM) to all four hydrophobic pockets of HBc capsids' asymmetric unit (N = 1.05 ± 0.1) (*Figure 2B*, *Figure 2—figure supplement 1*). ITC of geraniol with HBc showed a slightly enhanced

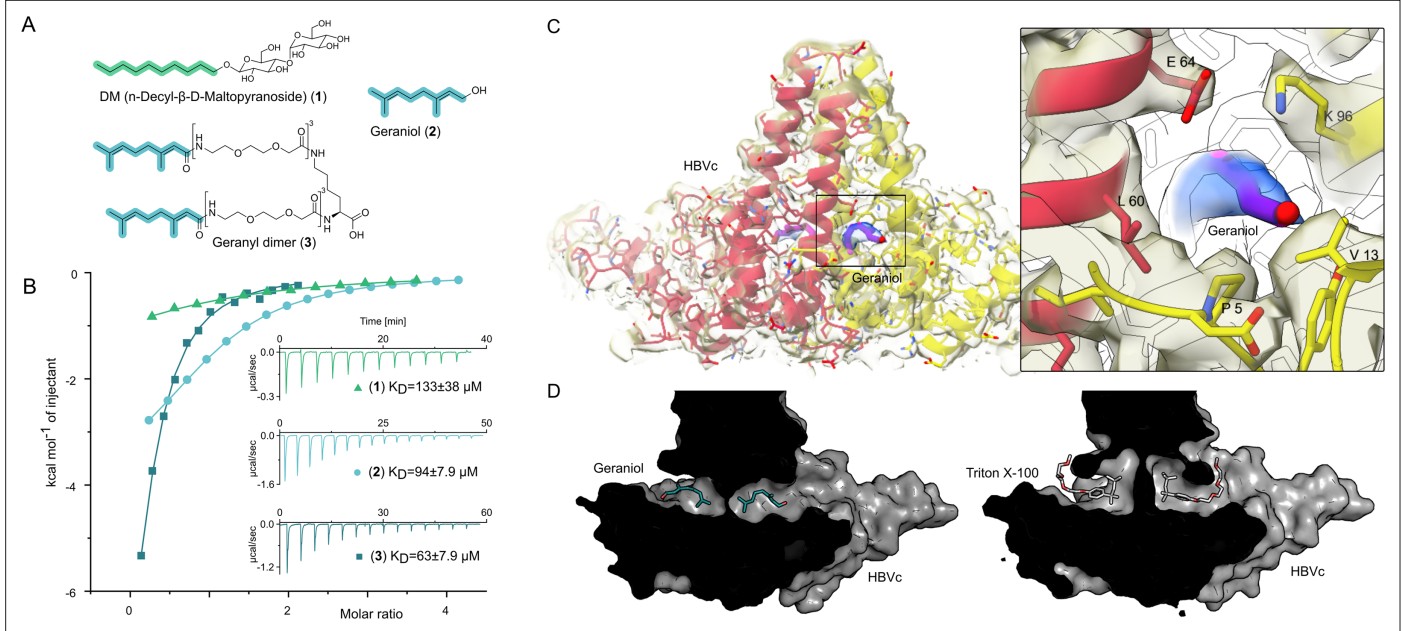

**Figure 2.** The central hydrophobic pocket of HBc-dimer is targeted by hydrophobic molecules containing isoprene units. (**A**) Structures of different substances used for the isothermal calorimetric titration (ITC) and cryo-EM experiments. N-Decyl-beta-d-maltopyranoside (DM) (**1**) and geraniol (**2**). Using geranic acid, we synthesized geranyl dimer (**3**), a dimeric binder forked by a lysine and having a linker of six dioxaoctanoic units. (**B**) Representative ITC heat signatures of DM (**1**), geraniol (**2**), and the geranyl dimer (**3**) with HBc capsids. Heat release is detected upon titration of the ligands to the HBc solution, indicating stoichiometric binding interaction. 4 mM geraniol (**2**) was titrated into a solution of 210 μM HBc. A solution of 2 mM geranyl dimer (**3**) was titrated into a solution 200 μM HBc. 1.6–2 mM solutions of DM (**1**) were titrated into solutions with 90, 100, and 150 μM HBc, respectively. The control experiments where geraniol, geranyl dimer, and DM were titrated into buffer are shown in *Figure 2—figure supplement 1*. Integrated heat signatures in kcal·mol⁻¹ plotted against the molar ratio of titrants to HBc. Binding isotherms (solid lines) were determined using a curve fitting procedure based on a one-site model. Among the ligands, the geranyl dimer has the strongest affinity to HBc, expectedly surpassing the monovalent geraniol by twofold. (**C**) Structure of the geraniol (magenta) within the HBc binding site (yellow and red) together with close-up view of the binding site with the EM-densities. Geraniol and residues (P5, L60, K96, E64, and V13) involved in hepatitis B virus's (HBV) envelopment with natural phenotypes are depicted in stick representation. The EM density of geraniol is shown in the zoom-out in blue. (**D**) Side-by-side comparison of the overlapping HBc geraniol and TX100 (*Makbul et al., 2021b*) binding sites suggests conformational flexibility and the ability of the hydrophobic pocket to accommodate larger hydrophobic molecules.

The online version of this article includes the following figure supplement(s) for figure 2:

**Figure supplement 1.** Control titrations of substances used for isothermal calorimetric titration (ITC) experiments.

**Figure supplement 2.** Geraniol binding mode to the hydrophobic pocket at the base of the capsid spike.

**Figure supplement 3.** Rational design of multivalent binders.

micromolar affinity ($K_D$ = 94 ± 8 μM) (*Figure 2A and B*, *Supplementary file 1*, *Supplementary file 2* and *Figure 2—figure supplement 1*) and a stoichiometry of N = 1.01 ± 0.04, implying that all four hydrophobic pockets of the asymmetric unit are occupied simultaneously. To confirm geraniol's binding to the capsids and resolve the molecular details of this interaction, we conducted cryo-EM of a mixture of HBc with excess of geraniol followed by single-particle analysis. This experiment resolved an additional density for geraniol in all four hydrophobic pockets within the asymmetric unit of HBc capsids (*Figure 2D*, *Figure 2—figure supplement 2*), confirming the thermodynamic binding data and further defining the underlying molecular interactions of the involved HBV residues P5, L60, K96, and F97.

Encouraged by structural confirmation of geraniol binding to the central hydrophobic pocket, we designed and synthesized a dimeric version of geraniol potentially capable of simultaneous binding to the HBc dimer. We connected the two geranyl moieties with a polyethylene glycol (PEG) linker that could bridge the distance of 38 Å between the two opposing hydrophobic pockets (see *Figure 2—figure supplement 3* for the design rationale). After synthesis, purification and mass spectrometric validation (Appendix 1) we determined the HBc capsid binding parameters of the geranyl dimer

via ITC. The analysis suggested that the dimer engages with both HBc binding sites simultaneously, resulting, however, only in a moderately enhanced micromolar affinity of 63 ± 8 μM (*Figure 2B*).

## Targeting the pocket of capsid spike tips with sub-micromolar affinity peptide dimers

Although geraniol and geranyl dimer displayed improved affinity to HBc and allowed structural insights on a binding pocket located at the center of HBc dimers, micromolar affinity is suboptimal for a functional compound. Therefore, we proceeded to explore another binding site located on the capsid spike tips formed by HBc dimers (*Figure 1*; *Böttcher et al., 1998*).

Earlier studies have shown that phage display-derived peptides were binding to the spike tips of recombinant HBc capsids. These peptides were also observed to disrupt the interaction between HBc and HBV's surface protein, L-HBs (*Wang et al., 1995*). Recently, we have shown that these peptides MHRSLLGRMKGA (P1), GSLLGRMKGA (P2), and the core binding motif SLLGRM bind to wild-type (wt) and mutant HBc variants (P5T, L60V and F97L) with intermediate micromolar $K_{D}$s of 26, 68, and 130 μM, respectively (*Makbul et al., 2021a*).

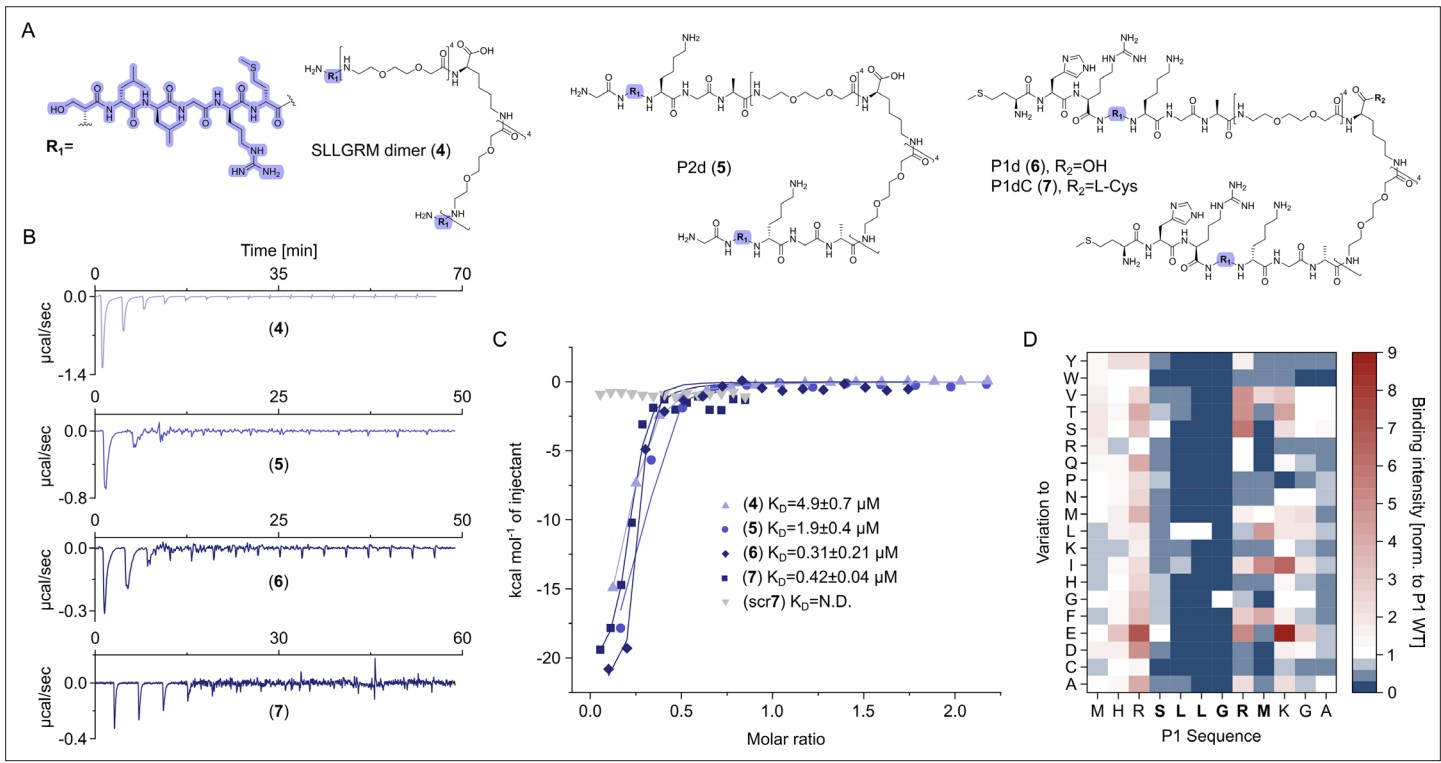

**Figure 3.** Dimeric peptide spike binders display strong low micromolar and sub-micromolar affinity. (**A**) Chemical structures of the dimeric peptides, all contain the core binding sequence -SLLGRM and share the same PEG linker and a lysine as the branching element of the dimer. (**B**) Exemplary isothermal calorimetric titration (ITC) thermograms showing the titration heat signature of HBc with dimers. A solution of 1500 μM (**4**) was titrated into a solution 150 μM HBc. A solution of 125 μM (**5**) was titrated into a solution 25 μM HBc. A solution of 200 μM (**6**) was titrated into a solution 25 μM HBc. A solution of 100 μM (**7**) was titrated into a solution 25 μM HBc. (**C**) The peptide dimers display low micromolar to sub-micromolar affinity to HBc, the affinity increases with the elongation of the binding sequence. (**D**) Sequence requirements of the HBc Spike binding site. Full positional scan of the P1 peptide sequence in microarray format, in which each residue was varied to each other proteogenic amino acid. Note that a drop in binding intensity upon variation of the core motif SLLGRM (highlighted in bold) substantiates its critical involvement in HBc binding. Refer to ***Supplementary file 3*** for the corresponding absolute grayscale values. Affinity gains observed for exchanging positively charged for negatively charged amino acids may be assay-specific false-positives as highlighted previously (*Makbul et al., 2021a*).

The online version of this article includes the following figure supplement(s) for figure 3:

**Figure supplement 1.** scrP1dC does not interact with HBc.

**Figure supplement 2.** Overview of equilibria between the asymmetric unit of HBc and peptide-binders.

**Figure supplement 3.** High turbidity is observed upon addition of HBc binding dimers to HBc solution.

Here, we designed dimeric peptides with a PEG linker capable of bridging the distance of 50 Å between the capsid spikes, thus tailoring our binders for simultaneous binding of two HBc dimers (*Figure 1B*, *Figure 2—figure supplement 3*). The three distinct dimeric peptides, the minimal SLLGRM dimer (**4**), the P2 dimer (P2d) (**5**), and two P1 dimers (P1d) (**6**) and P1dC (**7**), were synthesized, purified, and validated using mass spectrometry (Appendix 1). Subsequently, their binding to the HBV capsid was evaluated through ITC (*Figure 3A*, *Supplementary file 2*, *Figure 2—figure supplement 1*, *Figure 3—figure supplement 1*). With a $K_D$ value of 4.9 ± 0.7 µM, the SLLGRM-dimer (**4**) has the lowest affinity to HBc, followed by the P2-dimer (**5**) ($K_D$ = 1.9 ± 0.4 µM). Finally, the P1-dimers (**6**) and (**7**) displayed sub-micromolar affinities of 312 nM and 420 nM. Thus, P1-, P2-, and SLLGRM-dimers show 83-, 36-, and 27-fold increased affinities compared to their monomeric counterparts.

The significant increase in affinity of the P1-dimer over the monomer, by almost two orders of magnitude, may not be solely attributed to binding to two sites simultaneously. Once the P1-dimer binds, it can interact with up to four binding partners in its vicinity (*Figure 3—figure supplement 2*; *Wynne et al., 1999*). This may enable detachment and immediate reattachment to a nearby binding partner, further enhancing the local concentration and the overall binding strength of the P1 dimer.

Notably, while performing the ITC titrations we have noticed fast fluctuations of the heat signature baseline across the tested ligands (*Figure 3*, *Figure 3—figure supplement 1*), at least for the P1 dimer this phenomenon may be attributed to aggregation. To rule out any non-specific interactions caused by the PEG linker or handle for both the geranyl dimers as well as the P1/2 dimers, a scrambled dimeric peptide was used as a negative control. This scrambled peptide showed no detectable binding (*Figure 3C*), thereby confirming that the observed binding is specific to the designed peptide sequence and not influenced by the linker or other structural components.

To further substantiate and quantify a possible dimer-induced HBc aggregation, we next performed a turbidity assay (*Zhao et al., 2016*). We found that P1 dimer induces turbidity of a HBc solution already at 1:10 equivalents of HBc, whereas the P2 dimer was slightly less potent and the SLLGRM dimer did not induce turbidity at the same conditions and further required significantly higher concentrations

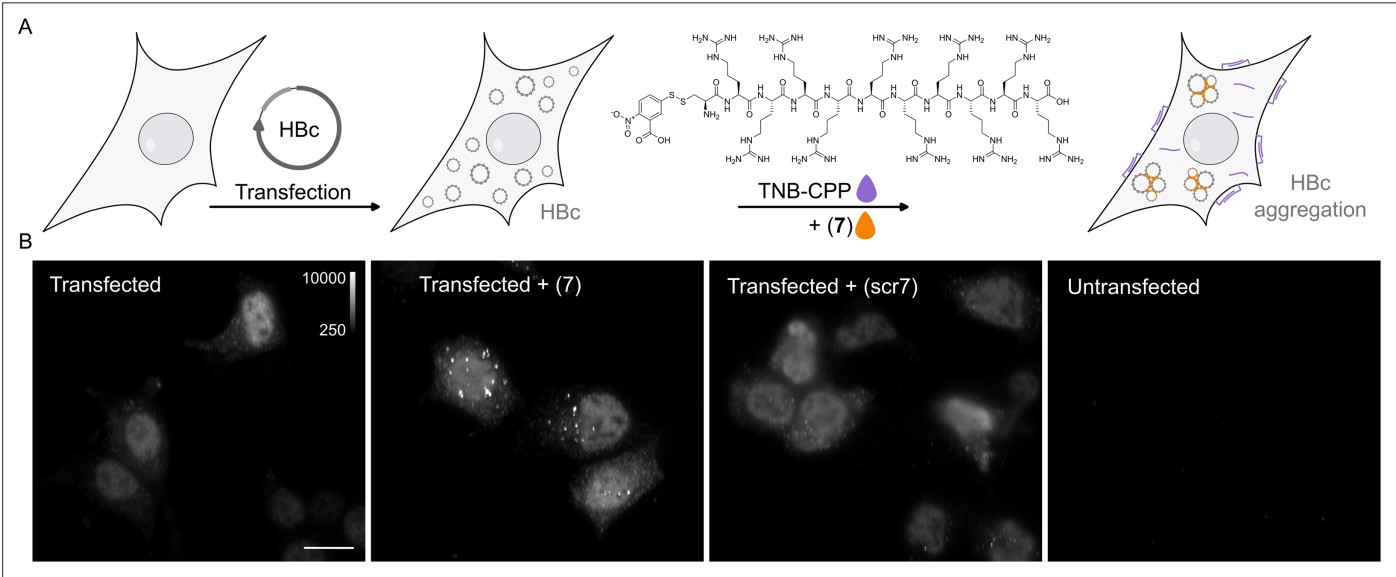

**Figure 4.** P1dC aggregates HBc in living HEK293 cells. (**A**) A polyarginine cell-penetrating peptide containing a cysteine with a TNB-activated thiol (gray highlight (**8**)). (**B**) The live cell experiment flow. First, mammalian cells are transfected with HBc coding plasmid. Then, after the cells express the protein, a mix of (**8**) and (**7**) is applied. The excess CPP facilitates membrane permeation, allowing (**7**) to enter the cell after a brief incubation. Once inside, (**7**) is separated from the CPP and can interact with the capsids. (**C**) After 1 hour incubation with (**7**) or scr(**7**), the cells were immediately washed, fixed, and labeled with anti HBc mAb16988 and a secondary DyLight650 conjugated antibody. The cells were visualized on wide-field fluorescent microscope with identical conditions and are presented with the same grayscale range. Transfected and untreated cells display diffuse HBc distribution, with clear fluorescence at the nucleus. Transfected cells treated with (**7**) display bright aggregates, whereas transfected cells treated with scr(**7**) have similar diffuse labeling as the untreated cells. Non-transfected cells are non-fluorescent. Scale bar 20 µm.

The online version of this article includes the following figure supplement(s) for figure 4:

**Figure supplement 1.** HBc aggregates appear after treatment with P1dC.

ratio relative to HBc (*Figure 3—figure supplement 3*). To shed light on the seemingly sequence-specific aggregation properties of the different dimers, we analyzed the binding of 240-point mutated P1 peptide variants in array format (*Figure 3D*, *Supplementary file 3*). The analysis recapitulated our earlier resolved sequence requirement for HBc binding and substantiated that the minimal sequence SLLGRM is the major mediator of HBc binding. Importantly, it further indicates that the additional N-terminal residues in P1 sequence are neither conserved nor critically required for binding despite their importance in inducing HBc aggregation.

## P1dC aggregates HBc in living HEK293 cells

The sub-micromolar affinity of the P1-dimer, along with its ability to induce capsid aggregation in vitro, prompted us to evaluate its effect on HBV core protein in living cells. To adapt the peptide for the intracellular delivery, we synthesized a C-terminally cysteinated version of P1-dimer, P1dC (**7**) (*Figure 3A*), and its scrambled counterpart scrP1dC scr(**7**), as well as a thiol-reactive polyarginine-based cell penetrating peptide (CPP), containing a cysteine, with a 5-thio-2-nitrobenzoic acid (TNB)-modified thiol (*Figure 4A*, Appendix 1). At the core of this intracellular delivery method is the in situ conjugation of the cargo molecule to a molar excess of a CPP via a disulfide bond, and the application of this reaction mix on living cells. The excess of the CPP over the active compound enable the reaction of CPP-thiols with the cellular surface, facilitating the penetration of the cargo-CPP conjugate. In turn, the disulfide bond between CPP and the cargo is reduced in the cytosol, separating the cargo from the CPP, allowing unhindered activity of the cargo molecule within the cell (*Figure 4B*; *Schneider et al., 2021a*; *Schneider et al., 2021b*).

To verify that the P1dC performs similarly to P1-dimer,, we performed another ITC assay to determine the affinity of the compound to HBc. The ITC confirmed that P1dC has an affinity of 420 ± 38 nM, comparable to P1-dimer, while the scrambled peptide did not display binding to HBc (*Figure 3*, *Figure 3—figure supplement 1*, *Supplementary file 2*). Thereafter we transfected mammalian cells (HEK293) with a plasmid coding for HBc. The cells expressed the protein for 2 days and were then treated for 1 hour with the thiol-activated cell penetrating peptide and P1dC or the respective negative control peptide scrP1dC (*Figure 4B*). Afterward, the cells were immediately washed and fixed and HBc was visualized with anti-HBc antibody and a secondary DyLight650 conjugated antibody. Transfected but otherwise untreated cells showed a homogeneous distribution of recombinant HBc molecules in the nucleus and to a lesser extent in the cytoplasm (*Figure 4C*). Yet, upon administration of 10 µM of P1dC (**7**), we observed aggregates of HBc (in the form of large bright spots) within the cells (*Figure 4C*, *Figure 4—figure supplement 1*). At a concentration of 10 µM, the scrambled dimer scrP1dC did not induce aggregation and the distribution of HBc remained largely homogenous.

Our live cell experiments have corroborated our in vitro findings, providing us a visual proof of P1dC-mediated HBc aggregation in a living cell. Thus, the peptide dimer causes an aggregation that resembles the HAP induced aggregation of the core protein and, like CAMs, can be expected to have the potential to disrupt the HBV life cycle.

## Cryo-EM confirms peptide-induced HBc aggregation

To affirm the capsid-aggregation property of our peptide dimers, we incubated solubilized purified capsid-like particles (CLPs, spherical capsid-like HBc multimers purified from *Escherichia coli*) with an excess of SLLGRM-dimer or P1dC, applied them on carbon grids, and imaged them using cryo-EM. The effect of peptide dimers on CLPs was already seen on the microscale cryo-EM images (*Figure 5—figure supplements 1 and 2*), with P1dC inducing large protein aggregates with multi-micron diameter. The less potent SLLGRM-dimer also induced visible aggregation, although with smaller aggregate size, while geraniol-treated samples showed minimal aggregation, and the smallest observed aggregate sizes. In the nanoscale, we observed clumped CLPs (*Figure 5—figure supplement 3*) and resolved the binding of both peptide dimers to the spike tips (*Figure 5*, *Figure 5—figure supplement 3*). The densities corresponding to bound peptide-dimers in both EM-reconstructions have volumes, which can accommodate a peptide chain of approximately six amino acid residues (*Figure 5*, *Figure 5—figure supplement 3*).

The asymmetric unit of HBc capsids (T = 4) is a tetramer consisting of an A/B- and C/D-dimer, which have slightly different 3D structures. Interestingly, the SLLGRM-dimer binds the A/B-dimer as well as the C/D-dimer, in contrast to the monomeric SLLGRM, which binds only to the tip of the

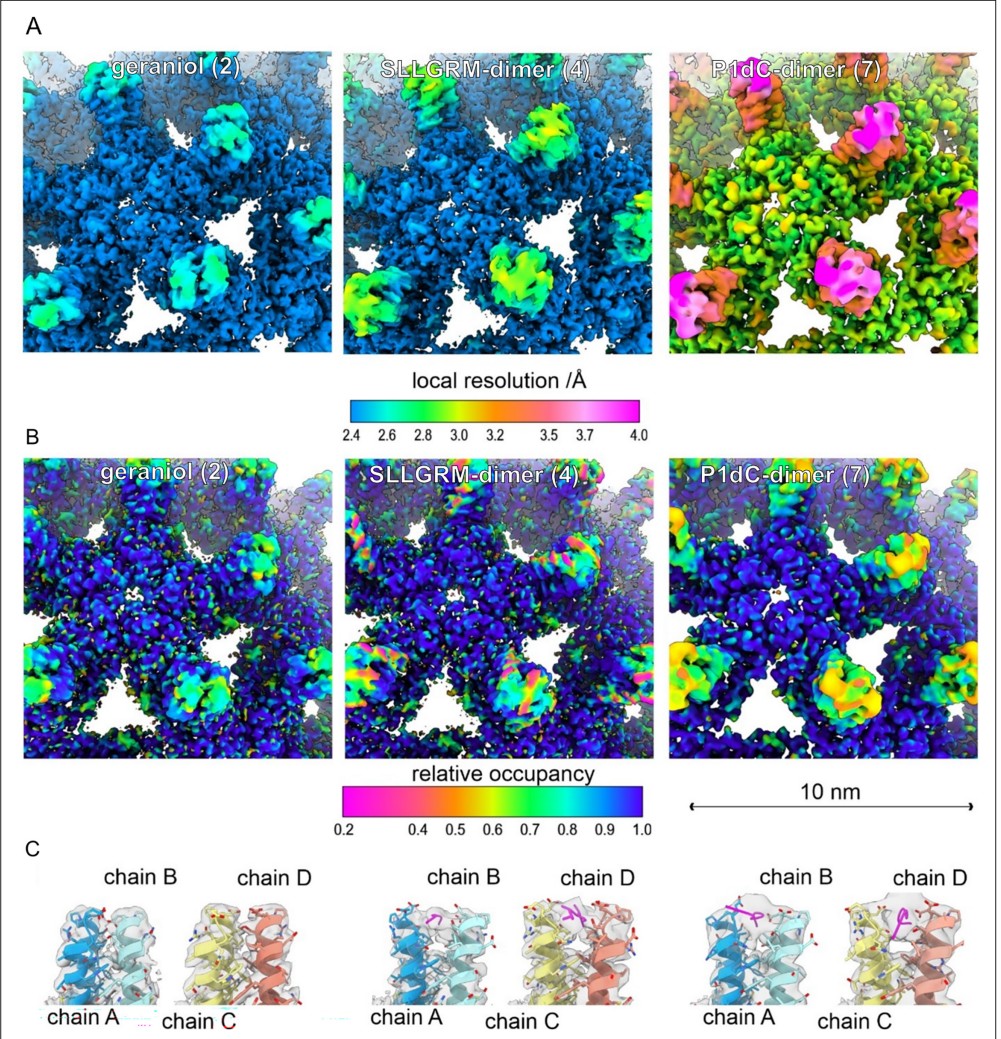

**Figure 5.** Peptide dimers binding the spike tips. Close-up of the surface representation of the EM-maps of capsid-like particle (CLP) incubated with geraniol (**2**), with SLLGRM-dimer (**4**) and P1dC (**7**). (**A**) The surface of the map is colored according to the local resolution. The map of (**7**) has a lower overall resolution, which is consistent with the lower number of particles in the reconstruction (***Supplementary file 4***). In all three maps the tips of the spikes are less well resolved than the capsid shell regardless of whether peptides are bound or not. This is in line with the general flexibility of the protruding spikes in HBc-CLPs (***Böttcher et al., 1998***; ***Hadden et al., 2018***). (**B**) The surface of the maps is colored according to the relative occupancy based on the gray value distribution as determined with OccuPy (***Forsberg et al., 2023***). Low relative occupancy cannot be distinguished from local flexibility. As the tips of the spike are flexible, they show generally lower occupancy than the protein shell. Comparing the relative occupancies in samples incubated with (**4**) and (**7**) suggests a lower occupancy with (**4**) than with (**7**). (C) Fit between the model and the map (gray, translucent) at the tips of spikes. Binding of an (**4**) or of (**7**) splays the helices at the tips apart similar as previously reported for binding of a P2-monomer (***Makbul et al., 2021a***). (**4**) binds to both quasi equivalent sites in contrast to SLLGRM-monomers, which binds only to the CD-dimer and does not show such a prominent splaying (***Makbul et al., 2021a***). Geraniol binds at the center of the spikes and does not change the conformation at the tips of the spikes.

The online version of this article includes the following figure supplement(s) for figure 5:

**Figure supplement 1.** Electron microscopy and image processing of HBc-CLPs with binders.

**Figure supplement 2.** Cryo-EM confirms strong capsid aggregation with peptide dimers.

**Figure supplement 3.** Nanoscale resolution of the dimer binding sites by cryo-EM.

C/D-dimer (*Makbul et al., 2021a*; *Figure 5*), in line with the multivalent binding and the higher affinity we measured with ITC. A possible mode of action of the peptide-dimer-induced aggregation is shown in *Figure 1B*.

The live cell experiments showed the formation of HBc aggregates upon incubation with P1dC. Yet, in live cells HBc may exist as a monomer, a dimer, a multimer, or a whole capsid; therefore, the observed aggregates were not necessarily formed by whole capsids. The cryo-EM experiment, however, provides confirmation that the peptide dimers have the capability to interact with complete CLPs, an important feature, which implies that the dimers have the potential to affect intact capsids upon cell infection.

## Discussion

In this study, we focused on the HBV core protein, a protein essential for HBV proliferation and virulence. We explored the druggability of two alternative, non-HAP, binding pockets on the HBc ultrastructure and developed synthetic dimers that target these pockets with sub-micromolar affinity, resulting in the aggregation of HBc.

DM and geraniol were selected as the water-soluble mimetics of the natural post-translational modifications of HBV/HDV components. We have demonstrated their ability to interact with the vital central hydrophobic pocket of HBV and showed a binding affinity improvement upon dimerization of the geraniol ligand. Higher affinity ligands may be developed into even more potent binders of the hydrophobic pocket using the outlined linker design, potentially exerting a pharmacological effect on HBV (*Briday et al., 2022*).

Earlier works demonstrated that point mutations within the HBc can significantly affect HBV infectivity, particularly through disruptions in HBc interactions (*Yuan et al., 1999b*; *Bruss, 2007*; *Ponsel and Bruss, 2003*; *Koschel et al., 1999*). These mutations at the base of the spike and the groove between spikes highlight the importance of these regions in viral replication. Using our structural knowledge of the capsid, particularly the distances between the spikes, we designed peptide dimers with the ability to simultaneously bind to neighboring spikes on the same capsid or attach to two distinct capsids (*Figure 3—figure supplement 2*). Our in vitro assays demonstrated that these peptide dimers display a robust affinity ranging from low micromolar to sub-micromolar levels (*Figure 3* and *Supplementary file 2*). Specifically, the peptide dimer (P1dC) with sub-micromolar affinity ($K_D$ = 420 ± 40 nM) is a promising candidate for a lead molecule for new a new class of CAMs. The peptide dimer, but not its scrambled dimeric counterpart, induced HBc aggregation in live mammalian cells expressing HBc. An effect resembling the aggregation was observed after a treatment with the classical CAM HAP (*Wu et al., 2013*).

An intriguing possibility would be that the spike and hydrophobic pocket interact or influence each other when binding different ligands. Molecular dynamics simulations reveal notable flexibility in HBV capsids, suggesting that structural asymmetry might impact ligand binding. Structural analysis of the HBV capsid shows that P2 peptide binding to the capsid spikes increases flexibility, while exerting a minimal impact on the underlying hydrophobic pocket. However, TX100 binding within the hydrophobic pocket influences the spike tips by flipping Phe97, whereas geraniols bound within the same pocket do not induce this change, indicating that this interaction is ligand-specific. Nevertheless, simultaneous application of spike binders and hydrophobic pocket ligands that modify spike conformation may offer valuable structural and functional insights into HBV capsids.

An intriguing possibility would be that the spike and the hydrophobic pocket could interact or exert an effect on each other upon binding various ligands. Molecular dynamics simulations revealed significant flexibility of the HBV capsids and suggested that structural asymmetry may affect ligand binding (*Pavlova et al., 2018*; *Perilla et al., 2016*; *Pavlova et al., 2022*). HBV capsid structural analysis showed that P2 peptide binding to the capsid spikes increases flexibility (*Makbul et al., 2021a*) while exerting a minimal impact on the hydrophobic pocket beneath. However, TX100 binding to the hydrophobic pocket affected the spike tips by flipping Phe97 (*Makbul et al., 2021b*) in the pocket, but the geraniols resolved within the hydrophobic pocket did not flip Phe97, thus suggesting that this cross-talk is ligand-specific. Nevertheless, a simultaneous application of spike binders and hydrophobic pocket ligands able to affect the spike conformation may provide valuable structural and functional insights into HBV capsids.

While our results are highly encouraging, application in complex organisms may require alternative delivery methods, investigation of HBV proliferation in infection models, and further study of immunogenicity and stability. Future studies should thoroughly assess the cytotoxic potential of peptide-induced HBc aggregation to determine any adverse effects at the cellular level, which will be crucial for evaluating the therapeutic potential of these compounds. Long-term cytotoxicity studies on cellular viability are essential to optimize these binders for clinical applications. Although our study targets two ligand-binding pockets on the capsid surface, a direct effect on HBV infectivity remains to be demonstrated. Prior mutational data, however, suggest that even minor perturbations, such as ligand binding, could mimic deleterious mutations and impair viral function. This study's insights into the unexplored pharmacological potential of these binding pockets and the compounds targeting them may lead to the development of new agents that impact viral capsids. Unlike classical CAMs, the peptide dimers exhibit a different mechanism of action and may act synergistically with CAMs or other antivirals. Further biological investigations will clarify the antiviral potential, applicability, and potency of compounds targeting the non-HAP binding pockets.

# Materials and methods

## Key resources table

| Reagent type (species) or resource | Designation | Source or reference | Identifiers | Additional information |
|---|---|---|---|---|
| Gene (hepatitis B virus genome) | CLP coding region; complement (733.1371) | GenBank: V01460.1 | | Genotype D; strain ayw |
| Cell line (Homo sapiens) | HEK293 | DSMZ | Cat# ACC 305 RRID:VCL_0045 | Epithelial morphology; embryo kidney |
| Transfected construct (human) | pEGFP-C2 | Clontech | | Donor vector for CLP expression |
| Antibody | Mouse anti-Hepatitis B Virus Antibody; core Antigen; clone C1-5, monoclonal; a.a. 74–89 | MilliporeSigma | Cat# MAB16988 RRID:AB_11212378 | IF (1:500) |
| Antibody | Goat anti-Mouse IgG (H+L), polyclonal | Invitrogen | Cat# 84545 RRID:AB_2633280 | IF (1:500) |
| Chemical compound, drug | n-Decyl-β-D-maltopyranoside (DM) | Anatrace | Cat# D310 | Used in binding assays. |
| Software algorithm | Relion 3.1. and 4.0 | https://github.com/3dem/relion; *Scheres, 2012a*; *Scheres, 2012b* | | Image processing |
| Software, algorithm | Cryosparc 4.0 | https://Cryosparc.com/ | | Image processing |
| Software, algorithm | MotionCorr2 | *Zheng et al., 2017* | | Movie processing |
| Software, algorithm | Phenix | https://phenix-online.org/ | | Model building |
| Software, algorithm | Coot | https://www2.mrc-lmb.cam.ac.uk/personal/pemsley/coot/binaries/ | | Model building |
| Software, algorithm | Chimera | https://www.rbvi.ucsf.edu/chimera | | Preparation of figures from pdb models and EM-maps |
| Software, algorithm | MicroCal ITC200 Analysis | Malvern Panalytical, Malvern | | Processing of the ITC data implemented in Origin and supplied with the MicroCal iTC200 |

Unless otherwise noted, all resins and reagents were purchased from IRIS biotechnologies or Carl Roth and used without further purification. All solvents were HPLC grade. All water-sensitive reactions were performed in anhydrous solvents under positive pressure of argon.

## Peptide synthesis

The peptides were produced using standard solid-phase peptide synthesis with Fmoc chemistry. Shortly, 2-chlorotrityl resin (1.6 mmol/g) was swollen in dry dichloromethane (DCM) for 30 minutes, then, the

desired amino acid (AA) (1eq) and the Boc-Gly-OH (1eq) with 4 eq. of dry N,N-diisopropylethylamine (DIEA) were added to the resin slurry. After overnight (ON) reaction at room temperature (RT) with agitation, the resin was capped with MeOH and washed with DCM and dimethylformamide (DMF). For the synthesis of cysteinated peptides, a 1% divinylbenzene Wang resin, preloaded with a 9-fluorenylmethyloxycarbonyl-Cysteine(Trityl)-OH [Fmoc-Cys(Trt)-OH] (0.4 mmol/g), was swollen in DMF for 30 minutes. Then, regardless of the resin type, Fmoc was removed using 20% piperidine in DMF solution and the resin was washed with DMF and DCM. After washes, the peptide chain was elongated by adding the desired amino acid (AA, 3 eq.) with ethylcyanohydroxyiminoacetate (Oxyma, 3 eq.) and N,N'-diisopropylcarbodiimide (DIC, 3 eq.). Capping was done with DIEA (50 eq.) and acetic anhydride (50 eq.) in N-methyl-2-pyrrolidone for 30 minutes. Coupling efficiency was monitored by measuring the absorption of the dibenzofulvene–piperidine adduct after deprotection. The peptide chain was elongated with further identical deprotection-conjugation cycles and after the completion the peptides were cleaved from the resin using a cocktail of 94% trifluoracetic acid (TFA), 3% $H_2O$, 3% triisopropylsilane (TIPS), for 4 hours at RT. The peptides were precipitated in ice-cold ether and then purified with semi-preparative HPLC and analyzed by LC-MS, as described below.

## Geranyl-dimer synthesis

2-Chlorotrityl resin (1.6 mmol/g) was swollen in dry DCM for 30 minutes. Then, 4 equivalents of Fmoc-Lys(Fmoc)-OH and 8 equivalents of dry DIEA were added to the resin slurry. The reaction was carried out ON at RT with agitation. After completion, the resin was capped with MeOH and washed with DCM and DMF. Fmoc deprotection was performed using a 20% piperidine solution in DMF, followed by thorough washing with DMF and DCM. The linker chain was then elongated by coupling 8-(9-fluorenylmethyloxycarbonyl-amino)–3,6-dioxaoctanoic acid (Fmoc-O2Oc-OH, 6 eq.) with Oxyma (6 eq.) and DIC (6 eq.). Capping was performed using 50 equivalents of DIEA and acetic anhydride in N-methyl-2-pyrrolidone for 30 minutes. Coupling efficiency was monitored by measuring the absorption of the dibenzofulvene–piperidine adduct after deprotection. The linker chain was further extended through two additional deprotection-conjugation cycles with Fmoc-O2Oc-OH. Subsequently, conjugation with geranic acid was carried out under similar conditions. The resulting dimer was cleaved from the resin using a 20% hexafluoroisopropanol solution in DCM. The solvents were removed via rotary evaporation, and the compound was purified using semi-preparative HPLC and analyzed by LC-MS, as described below.

## 5-(thio)-2-nitrobenzoate conjugation to the thiolated cell penetrating peptide (CPP)

A 10-mer oligoarginine peptide connected to a cysteine (C-RRRRRRRRRR) was reacted with 10 equivalents of 5,5-dithio-bis-(2-nitrobenzoic acid) in 1:1 DMF: 0.1 M phosphate buffer for 30 minutes with agitation at RT. Then the reaction mixture was directly injected in semi-preparative HPLC, purified, and analyzed by LC-MS, as described below.

## Purification and characterization of peptide-based probes

The compounds were purified from the crude reaction mix by reverse-phase HPLC using a water acetonitrile gradient with 0.1% formic acid (FA). Semi-preparative HPLC was performed on Shimadzu Prominence equipped with a diode-array detector (DAD) system using a C18 reverse-phase column (Phenomenex Onyx Monolithic HD-C18 100×4.6 mm or Onyx Monolithic C18 100×10 mm). Purity and structural identity were verified using a DAD equipped 1260 Infinity II HPLC with a C18 reverse-phase column (Onyx Monolithic C18 50×2 mm), coupled to a mass selective detector single quadruple system (Agilent Technologies). Compounds analyzed in ESI+ mode were run in a water-acetonitrile gradient with 0.1% FA. Compounds analyzed in ESI- mode were run in a 10 mM pH = 7 ammonium bicarbonate – acetonitrile gradient.

## Protein expression and purification HBc CLPs

The expression and purification of CLPs were done as previously described (*Makbul et al., 2021b*). Shortly, the recombinant HBV core protein (HBc) was overexpressed in *E. coli* (BL-21) and formed CLPs. CLPs were purified by fractionated ammonium sulfate precipitation followed by sucrose density

gradient centrifugation. The major capsid type (ca. 95%) was formed by 240 subunits (Triangulation: T = 4).

## Isothermal titration calorimetry (ITC)

Samples (ca. 8 mL) of purified capsids were filtered (Rotilabo syringe filter with a pore size of 220 nm, Carl Roth GmbH Co. KG, Karlsruhe, Germany), dialyzed against 1.4 L buffer A (40 mM HEPES, 200 mM NaCl, 1 mM $MgCl_2$, 1 mM $CaCl_2$, pH 7.5) using a dialysis membrane tube (Spectra Por Biotech cellulose ester tube, 1 MDa MWCO, Spectrum Laboratories, Inc, Rancho Dominguez, CA, USA). The dialysis was performed at 4°C under gentle stirring for 16 hours ON. The next day, the dialyzed sample was removed from the dialysis tube and concentrated in a centrifuge using a concentrator (30 kDa MWCO Spin-X UF 6 mL, Corning Inc, Corning, NY, USA). The concentrate was filtered (centrifugal filter unit Ultrafree MC, pore size of 100 nm, Merck KGaA, Darmstadt, Germany) and the concentration determined by the Bradford assay (Roti Nanoquant, Carl Roth GmbH Co. KG).

The peptide dimers were dissolved in the buffer from the dialysis of the capsids. In this buffer, SLLGRM dimer and P2 dimer have solubilities of at >8 mM and the P1 dimer of >2 mM. 4 mM geraniol was titrated into a solution of 210 µM HBc.

A solution of 2 mM geranyl dimer was titrated into a solution 200 µM HBc. 1.6–2 mM solutions of DM were titrated into solutions with 90, 100, and 150 µM HBc, respectively.

Before filling the ITC cell and syringe, all samples were degassed for 10 minutes at 20°C (ThermoVac, Malvern Panalytical, Malvern, Worcestershire, UK). Solutions of peptide dimers were titrated into solutions of capsids using a MicroCal iTC200 instrument (Malvern Panalytical) according to the specifications in *Supplementary file 1*. The resulting thermograms and isotherms were processed and fitted using the Origin software supplied with the iTC200 instrument. The thermograms were integrated and the corresponding isotherms were fitted using a one-site model. The peptide and geranyl dimers are bivalent and have 120 or 240 potential binding sites on CLPs, respectively. The two binding sites of peptide and geranyl dimers are not identical but very similar. This also true for the binding sites on capsids, so the binding energetics of the dimers are very similar and are best represented by a one-site model. All obtained thermodynamic parameters refer to concentrations of monomeric HBc. All ITC experiments were complemented with control experiments where solutions of peptide dimers were titrated into the dialysis buffer.

## Turbidity assay

All peptides were dissolved in buffer A (40 mM HEPES, 200 mM NaCl, 1 mM $MgCl_2$, 1 mM $CaCl_2$, pH 7.5) and the capsid solutions were filtered once (centrifugal filter unit Ultrafree MC, pore size of 100 nm, Merck KGaA). The concentrations of the P1, P2, and SLLGRM dimers were varied between 0.1 and 100 µM, and the concentration of HBc was kept constant at 10 µM for the P1 and P2 dimer and at 50 µM for the SLLGRM dimer. All experiments were performed using a standard photometer (GENESYS UV/VIS spectral photometer, Thermo Fisher Scientific, Hillsboro, OR, USA) at RT and at a wavelength of 350 nm using disposable UV transparent cuvettes (SARSTEDT AG & Co. KG, Sarstedtstraße 1, 51588 Nümbrecht/Germany).

## Cryogenic grid preparation of capsids in complex with peptide-dimers

In a plasma cleaner (model PDC-002. Harrick Plasma, Ithaca, NY, USA) holey carbon grids (R1.2/1.3, 300 mesh Cu grids, Quantifoil Micro Tools, Jena, Germany) were made hydrophilic by plasma cleaning. This was done at a pressure of 29 Pa for 2 minutes using ambient air as plasma medium at 'medium power' of the instrument. Solutions of purified HBc (200 µM) in complex with the P1dC- and the SLLGRM-dimer (each 400 µM) were prepared in buffer A. After the end of ITC experiment with geraniol and HBc (*Figure 2*), a sample from the cell of the ITC instrument was retrieved and used for freezing grids. 3.5 µL aliquots of each sample were applied onto the grids. For plunge freezing of grids, ethane was used as medium (liquefied by liquid nitrogen) with the help of a Vitrobot (mark IV, FEI Company, Hillsboro, OR, USA) using Whatman filter papers (type 541). The Vitrobot had the following settings: no wait and drain times, 6 s of blot time, blot force of 25 and a nominal humidity of 100%. The frozen grids were stored in liquid nitrogen for at least one night before being used for image acquisition.

## Cryo-EM and image processing

Cryo-EM was done as previously described (*Makbul et al., 2021b*). Shortly, movies were acquired with the software EPU on a Krios G3 electron microscope equipped with a Falcon III camera (Thermo Fisher Scientific) in integrating mode at a magnification of 75,000 with an accelerating voltage of 300 kV. The total exposure was 40 e$^-$/Å$^2$ and was fractionated over 20 fractions. For HBc CLPs with bound P1dC, three movies were acquired per hole and one hole was acquired per stage position. For HBc CLPs with bound SLLGRM dimers or bound geraniol, at each stage position three movies were acquired per hole from the central hole and from the four closest neighboring holes. The different movie positions at the same stage position were centered with image shift. Movies were motion corrected, exposure weighted, and averaged with MotionCorr2. *Figure 5—figure supplement 1A* shows representative corrected movie averages, which were imported to Relion for further processing. Each image shift position was treated as a different optics group in the subsequent image processing. Image processing was done with Relion 3.1 or Relion 4. As previously described (*Makbul et al., 2021b*), imposing icosahedral symmetry. At the end of the image processing with Relion (for CLPs with bound P1dC or SLLGRM-dimers), particle images were imported into CryoSparc 4.02 and were further refined with none uniform refinement (*Punjani et al., 2020*), including global and local CTF refinement and Ewald's sphere correction. Final maps were filtered with deepemhancer, or B-factor sharpened (CryoSPARC 'Sharpen' or 'relion_postprocess'). The resolution of the final maps was estimated by Fourier shell correlation (FSC = 0.143; after gold standard refinement) with 'relion_postprocess' (*Figure 5—figure supplement 1*). Parameters of the image acquisition and the processing are summarized in *Supplementary file 4*.

## Modeling of cryo-EM maps, refinement of PDB files and their validation

For modeling of the EM densities of HBc in complex with the peptide dimers, the PDB file 7od6 (*Makbul et al., 2021a*) was used as a starting model. This model represents the asymmetric unit of the HBc capsids with T = 4 packing. After slight modifications, the PDB model was fitted into the EM-map as a rigid body and refined iteratively using the software packages Coot (*Casañal et al., 2020*) and Phenix (*Liebschner et al., 2019*) and validated with MolProbity (*Prisant et al., 2020*). The resolution of the density at the tips of the capsids which we attributed to the binding segments of the peptide dimers was low. Therefore, these densities could only be modeled as poly-alanine chains. All figures showing EM-densities with or without the corresponding PDB models were prepared with Chimera (*Yang et al., 2012*).

## Cloning

Full-length wild-type (fl wt) HBc (genotype D; strain ayw; GenBank: V01460.1, MQLFHLCLIISCSCPT VQASKLCLGWLWGMDIDPYKEFGATVELSFLPSDFFPSVRDLLDTASALYREALESPEHCSPHHTALRQAIL CWGELMTLATWVGVNLEDPASRDLVVSYVNTNMGLKFRQLLWFHISCLTFGRETVIEYLVSFGVWIRTPPAY RPPNAPILSTLPETTVVRRRGRSPRRRTPSPRRRRSQSPRRRRSQSRESQC) (*Galibert et al., 1979*) was cloned into the pEGFP-C2 vector (Clontech) using Gibson assembly *Gibson, 2011* by replacing the gene sequence coding for eGFP. The vector and insert were amplified by PCR and purified by gel extraction (FastGene Gel/PCR Extraction Kit, Nippon Genetics Europe GmbH, Düren, Germany). The purified PCR products were assembled into a single plasmid construct using a home-made Gibson assembly reaction mixture. An aliquot of the reaction product was transformed into XL1 blue cells, plated onto LB-amp agar-plates, and grown at 37°C ON. Six colonies were used for the inoculation of 6 × 5 mL LB-amp medium. The cell cultures were grown under vigorous shaking in an incubator at 37°C ON. The next day, the plasmid DNA was extracted from the cell cultures (FastGene Plasmid Mini Kit, Nippon Genetics Europe GmbH) and sequenced by Sanger sequencing (Microsynth Seqlab GmbH, Göttingen, Germany). A plasmid construct containing the correct gene sequence of HBc was used for endotoxin-free plasmid DNA preparation (NucleoBond Xtra Midi EF, Macherey Nagel GmbH & Co. KG, Düren, Germany).

## HEK293 cell cultures and transfection

HEK293 cells were cultured in DMEM (Gibco), supplemented with GlutaMax and pyruvate (Gibco), 10% fetal bovine serum (FBS) (Gibco) and 1% Penicillin/Streptomycin (Sigma) at 37°C and with 5% $CO_2$. The cells were plated on 0.15-mm-thick 18 mm glass coverslips coated with 35 μg/mL poly-D-lysine in

a 12-well plate and were transfected with the cloned 1 µg plasmid DNA per coverslip using polyethylenimine (PEI). The transfection was performed at 60–80% confluence. Shortly before transfection, the medium was changed to fresh DMEM. The DNA was added to 100 µL DMEM without additives and mixed, 4 µL fresh PEI (1 mg/mL) was added, mixed immediately, and incubated for 20 minutes at RT. The transfection mix was pipetted drop-wise on cells while swirling and incubated ON. The medium was changed to fresh DMEM with 2% FBS after 12–24 hours, and on the following day the cells were used for live assays, then fixed and stained.

## Cell assays and immunocytochemistry

Live HEK293 cells expressing HBc were incubated in DMEM with 10 µM P1dC and with 10 µM of the scrambled version of the peptide, both peptides in situ activated with the reactive CPP. After 1 hour incubation at 37°C, the treated and untreated live HEK293 cells expressing HBc and the untransfected HEK293 cells were washed and fixed with 0.1 M sodium phosphate buffer pH 7.4 containing 4% paraformaldehyde (EM grade, Polysciences) and 1% sucrose for 10–20 minutes at 37°C. After three rinses in phosphate-buffered saline (PBS), the cells were permeabilized with 0.1% Triton X-100 in PBS for 10 minutes at RT, rinsed again and blocked for 1 hour in PBS with 3% bovine serum albumin. Then, primary mAb16988 (#6B9780, MilliporeSigma) and secondary DyLight650 (#84545, Invitrogen) antibodies were applied sequentially with 1:500 dilution in blocking solution for 1 hour.

## Wide-field fluorescence microscopy

The coverslips with the cell samples were inserted in an imaging chamber (Ludin Chamber Type 1, Life Imaging Services) and imaged in PBS. The measurements were taken from distinct samples with a sample size ≥2 for each group. A series of images, used to generate the datapoints, were acquired from different regions of the sample, each region having a distinct group of cells.

The samples were imaged on an inverted Leica DMI6000B microscope with a ×100 oil-immersion objective (NA 1.49) using a Leica DFC9000 GTC VSC-05760 sCMOS camera (16-bit, image pixel size: 130 nm). The 628/40 excitation and 692/40 emission filter was used for DyLight650, 10 images were acquired at a frame rate (exposure time) of 100 ms, and constant illumination intensity to ensure comparability (n≥10).

## Automated solid-phase peptide synthesis

µSPOT peptide arrays (*Dikmans et al., 2006*) were synthesized using a MultiPep RSi robot (CEM GmbH, Kamp-Lindford, Germany) on in-house produced, acid-labile, amino-functionalized, cellulose membrane discs containing 9-fluorenylmethyloxycarbonyl-β-alanine (Fmoc-β-Ala) linkers (average loading: 130 nmol/disc – 4 mm diameter). Synthesis was initiated by Fmoc deprotection using 20% piperidine (pip) in DMF followed by washing with DMF and ethanol (EtOH). Peptide chain elongation was achieved using a coupling solution consisting of preactivated amino acids (aas, 0.5 M) with ethyl 2-cyano-2-(hydroxyimino)acetate (oxyma, 1 M) and DIC (1 M) in DMF (1:1:1, aa:oxyma:DIC). Couplings were carried out for 3 × 30 min, followed by capping (4% acetic anhydride in DMF) and washes with DMF and EtOH. Synthesis was finalized by deprotection with 20% pip in DMF (2 × 4 µL/disc for 10 min each), followed by washing with DMF and EtOH. Dried discs were transferred to 96 deep-well blocks and treated, while shaking, with sidechain deprotection solution, consisting of 90% TFA, 2% DCM, 5% $H_2O$, and 3% TIPS (150 µL/well) for 1.5 hours at RT. Afterward, the deprotection solution was removed, and the discs were solubilized ON at RT, while shaking, using a solvation mixture containing 88.5% TFA, 4% trifluoromethanesulfonic acid (TFMSA), 5% $H_2O$, and 2.5% TIPS (250 µL/well). The resulting peptide-cellulose conjugates (PCCs) were precipitated with ice-cold ether (0.7 mL/well) and spun down at 2000 × g for 10 minutes at 4°C, followed by two additional washes of the formed pellet with ice-cold ether. The resulting pellets were dissolved in DMSO (250 µL/well) to give final stocks. PCC solutions were mixed 2:1 with saline-sodium citrate (SSC) buffer (150 mM NaCl, 15 mM trisodium citrate, pH 7.0) and transferred to a 384-well plate. For transfer of the PCC solutions to white-coated CelluSpot blank slides (76 × 26 mm, Intavis AG), a SlideSpotter (CEM GmbH) was used. After completion of the printing procedure, slides were left to dry ON.

## Peptide microarray-binding assay

The microarray slides were blocked for 60 minutes in 5% (w/v) skimmed milk powder (Carl Roth) PBS (137 mM NaCl, 2.7 mM KCl, 10 mM $Na_2HPO_4$, 1.8 mM $KH_2PO_4$, pH 7.4). After blocking, the slides were incubated for 15 minutes with 55 nM (monomer equivalent) of HBc in the blocking buffer, then washed 3× with PBS. HBc was detected with a primary 1:2500 diluted mAb16988 (anti-HBV antibody, core antigen, clone C1-5, aa 74–89, MilliporeSigma, Darmstadt, Germany) and a secondary 1:5000 diluted HRP-coupled anti-mouse antibody (31430, Invitrogen). The antibodies were applied in blocking buffer for 15 minutes, with three PBS washes between the antibodies and after applying the secondary antibody. The chemiluminescent readout was obtained using SuperSignal West Femto maximum sensitive substrate (Thermo Scientific GmbH, Schwerte, Germany) with a c400 Azure imaging system (lowest sensitivity, 90 s exposure time).

Binding intensities were quantified with FIJI (*Schindelin et al., 2012*) using the 'microarray profile' plugin (OptiNav Inc, Bellevue, WA, USA). The raw grayscale intensities for each position were obtained for the left and right sides of the internal duplicate on each microarray slide, n = 3 arrays in total. Blank spots were used to determine the average background grayscale value that was subtracted from the raw grayscale intensities of non-blank spots. Afterward, the spot intensities were normalized to the average grayscale value of the 14 replicates of peptide binder P1 ('MHRSLLGRMKGA').

## Acknowledgements

HMM acknowledges the support of the Junior Group Leader program of the Rudolf Virchow Center, University of Würzburg the excellent ideas programme of the JMU and support through DFG MA6957/1-1. BB acknowledges support for this project by the DFG (BO1150/17-1). Electron microscopic data were acquired at the cryo-EM facility in Würzburg funded by the German Research Foundation (DFG projects 359471283, 456578072, 525040890).

## Additional information

### Funding

| Funder | Grant reference number | Author |
|---|---|---|
| Deutsche Forschungsgemeinschaft | DFG MA6957/1-1 | Hans Michael Maric |
| Deutsche Forschungsgemeinschaft | BO1150/17-1 | Bettina Böttcher |
| Deutsche Forschungsgemeinschaft | 359471283 | Bettina Böttcher |
| Deutsche Forschungsgemeinschaft | 456578072 | Bettina Böttcher |
| Deutsche Forschungsgemeinschaft | 525040890 | Bettina Böttcher |

The funders had no role in study design, data collection and interpretation, or the decision to submit the work for publication.

### Author contributions

Vladimir Khayenko, Formal analysis, Investigation, Methodology, Writing – original draft, Writing – review and editing; Cihan Makbul, Formal analysis, Investigation, Methodology, Writing – review and editing; Clemens Schulte, Formal analysis, Investigation, Visualization, Methodology, Writing – review and editing; Naomi Hemmelmann, Formal analysis, Investigation, Writing – review and editing; Sonja Kachler, Validation, Investigation, Methodology; Bettina Böttcher, Data curation, Software, Supervision, Visualization, Project administration, Writing – review and editing; Hans Michael Maric, Conceptualization, Supervision, Writing – original draft, Project administration, Writing – review and editing

### Author ORCIDs

Bettina Böttcher ![ORCID] https://orcid.org/0000-0002-7962-4849
Hans Michael Maric ![ORCID] https://orcid.org/0000-0002-2719-4752

Reviewer #1 (Public review): https://doi.org/10.7554/eLife.98827.3.sa1
Reviewer #2 (Public review): https://doi.org/10.7554/eLife.98827.3.sa2
Author response https://doi.org/10.7554/eLife.98827.3.sa3

## Additional files

### Supplementary files

Supplementary file 1. ITC200 specifications. ITC200 instrument's specifications used for the interaction analysis between peptides and capsids.

Supplementary file 2. Thermodynamic parameters of HBc capsids interactions. Summary of thermodynamic parameters obtained by ITC experiments using the peptide dimers and fl wt HBc capsids. In case of the P2 dimer, the deviations represent deviations of fit since only one ITC experiment was performed. N represents stoichiometry.

Supplementary file 3. Microarray positional scan data. Obtained raw grayscale values from P1 full positional scan in µSPOT format. Each box corresponds to a single-point variation of the P1 peptide sequence (horizontal) as indicated in the first column. The raw intensity values presented here were used for calculating the fold intensity change of each point variation against the wildtype sequence. Data are presented as mean of n=3 microarray slides with SD.

Supplementary file 4. Cryo-EM data. Summary of cryo-EM data acquisition and image processing of the HBc CLPs with bound geraniol, P1dC, and SLLGRM-dimers.

MDAR checklist

### Data availability

Experimental Cryo-EM data acquisition and image processing data has been deposited in PDB and EMDB under accession codes: HBc CLPs + Geraniol (2), PDB 8PWO, EMD-17996; HBc CLP+ SLLGRM dimer, PDB 8PX6, EMD-18001; HBc CLP + P1dC, PDB: 8PX3, EMD-18000.

The following datasets were generated:

| Author(s) | Year | Dataset title | Dataset URL | Database and Identifier |
|---|---|---|---|---|
| Makbul C, Khayenko V, Maric MH, Bottcher B | 2024 | Hepatitis B core protein with bound SLLGRM-dimer | https://www.rcsb.org/structure/8PX6 | RCSB Protein Data Bank, 8PX6 |
| Makbul C, Khayenko V, Maric MH, Bottcher B | 2024 | Hepatitis B core protein with bound P1dC | https://www.rcsb.org/structure/8PX3 | RCSB Protein Data Bank, 8PX3 |
| Makbul C, Khayenko V, Maric MH, Bottcher B | 2024 | Hepatitis B core protein with bound Geraniol | https://www.rcsb.org/structure/8PWO | RCSB Protein Data Bank, 8PWO |
| Makbul C, Khayenko V, Maric MH, Bottcher B | 2024 | Hepatitis B core protein with bound Geraniol | https://www.ebi.ac.uk/emdb/EMD-17996 | Electron Microscopy Data Bank, EMD-17996 |
| Makbul C, Khayenko V, Maric MH, Bottcher B | 2024 | Hepatitis B core protein with bound SLLGRM-dimer | https://www.ebi.ac.uk/emdb/EMD-18001 | Electron Microscopy Data Bank, EMD-18001 |
| Makbul C, Khayenko V, Maric MH, Bottcher B | 2024 | Hepatitis B core protein with bound P1dC | https://www.ebi.ac.uk/emdb/EMD-18000 | Electron Microscopy Data Bank, EMD-18000 |

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

## Appendix 1

### Chromatographic and mass spectrometric analytical data

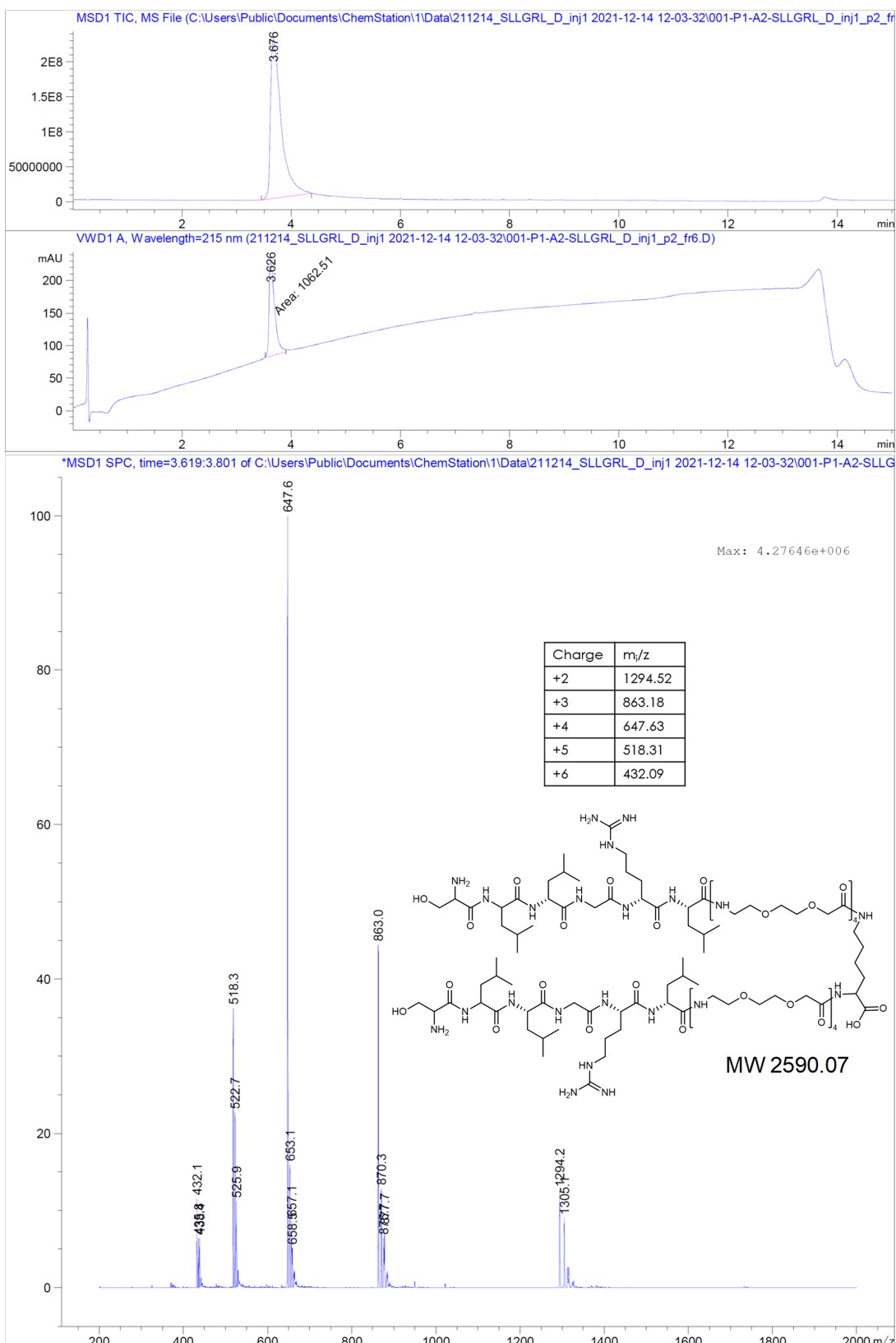

**Appendix 1—figure 1.** SLLGRL-PEGlinker-Dimer.

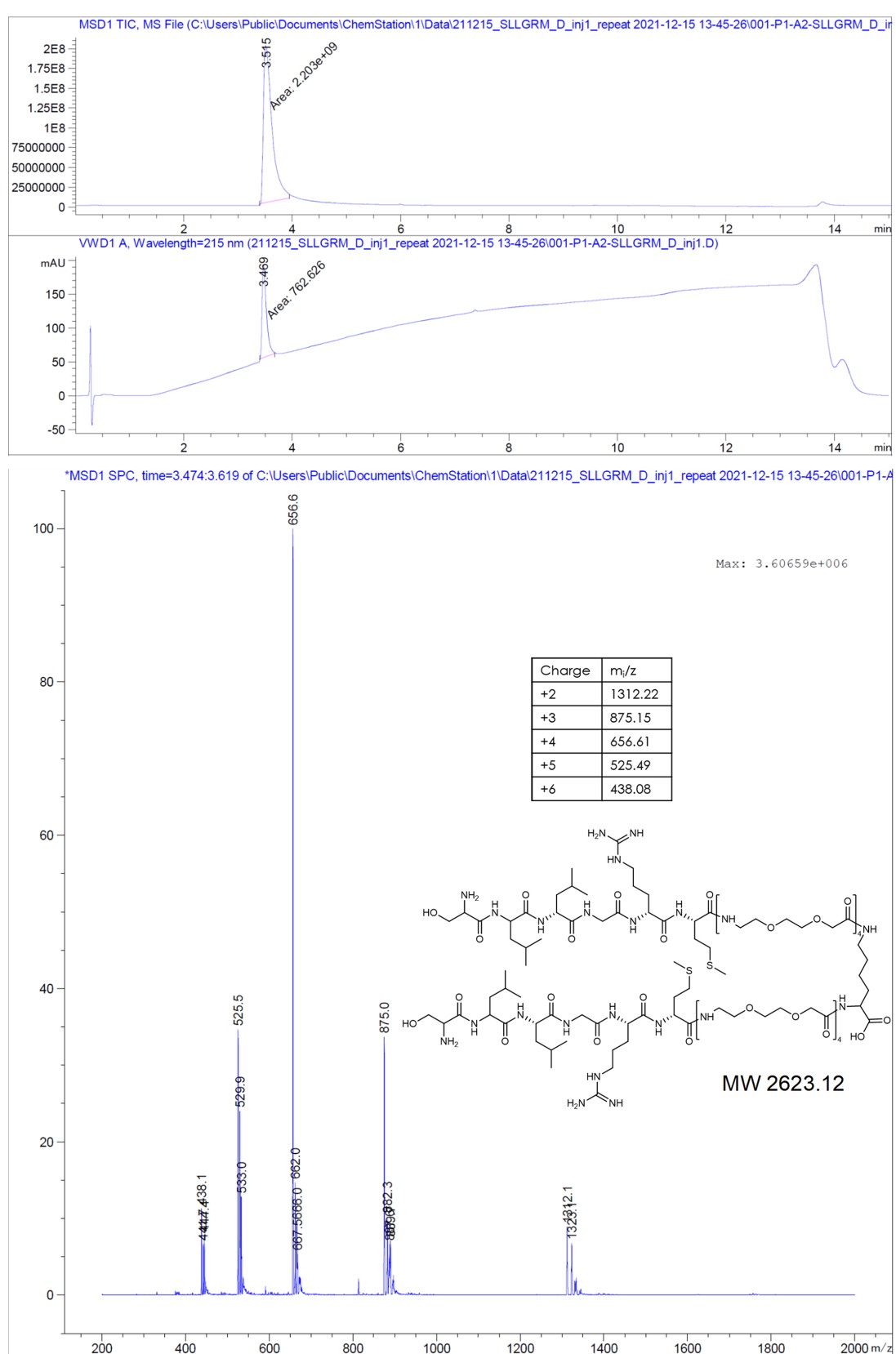

**Appendix 1—figure 2.** SLLGRM-PEGlinker-Dimer (SLLGRM dimer).

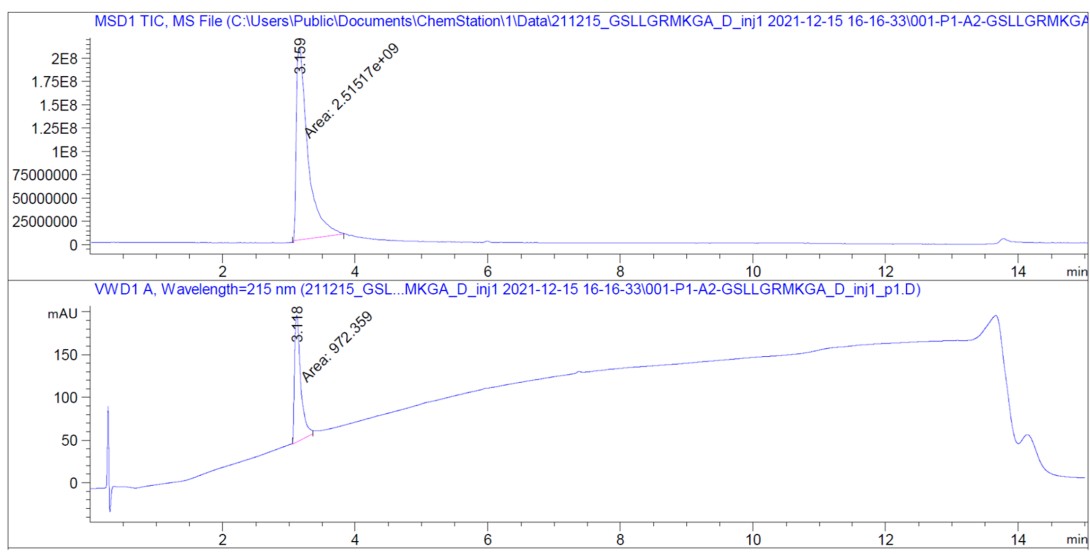

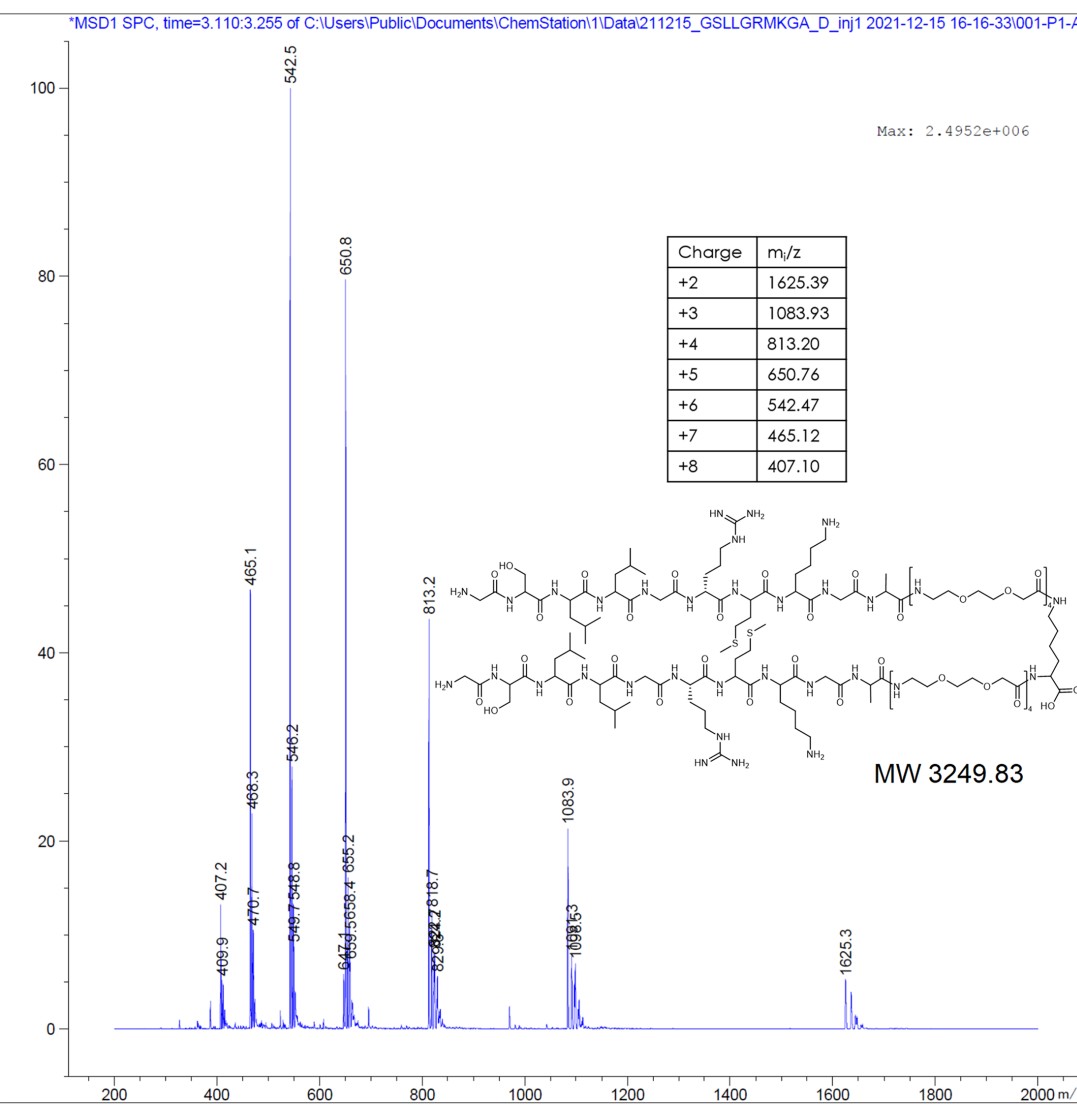

| Charge | m$_i$/z |
|--------|---------|
| +2 | 1625.39 |
| +3 | 1083.93 |
| +4 | 813.20 |
| +5 | 650.76 |
| +6 | 542.47 |
| +7 | 465.12 |
| +8 | 407.10 |

MW 3249.83

**Appendix 1—figure 3.** GSLLGRMKGA-PEGlinker-Dimer (P2 dimer).

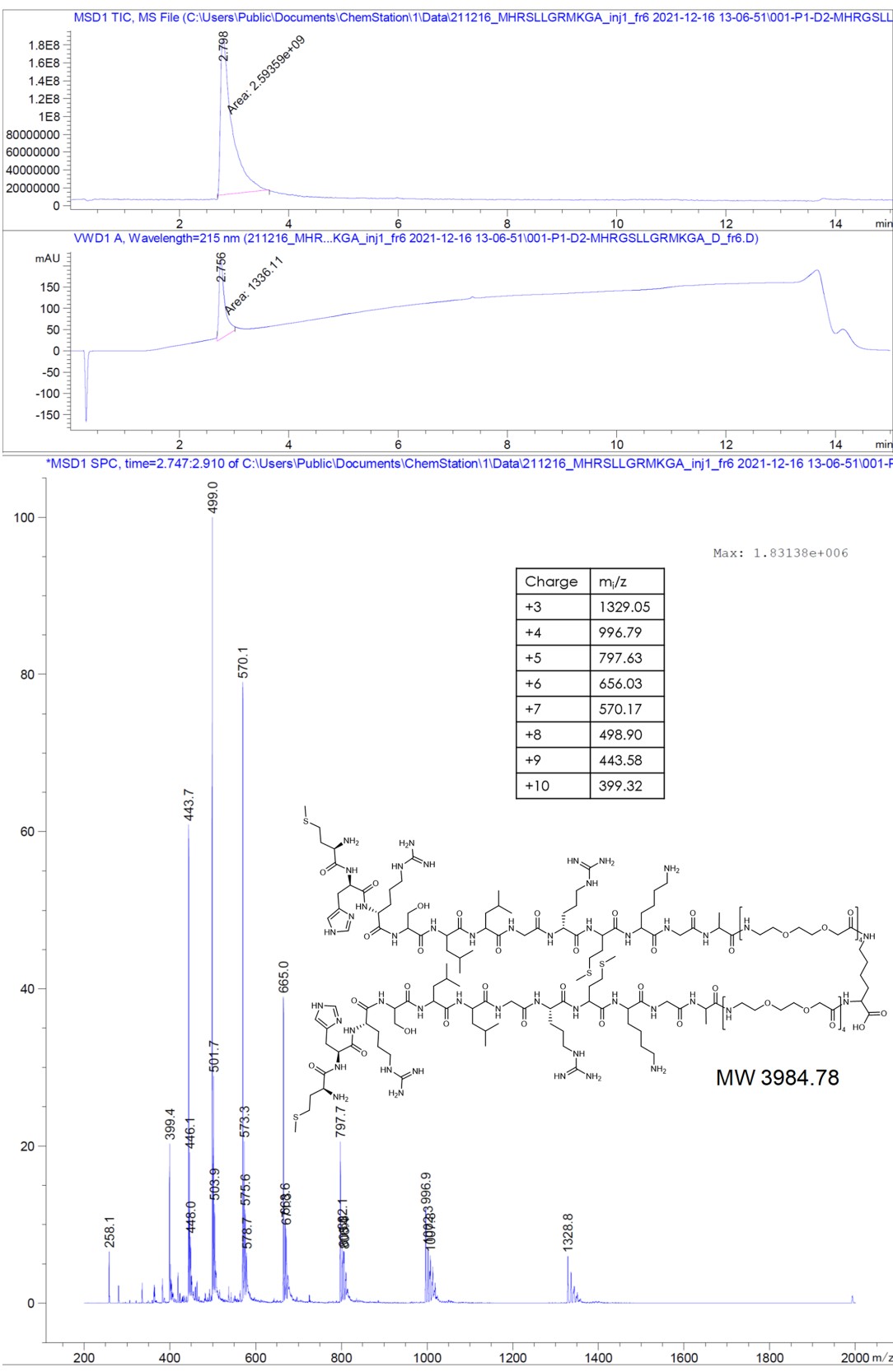

**Appendix 1—figure 4.** MHRSLLGRMKGA-PEGlinker-Dimer (P1 dimer).

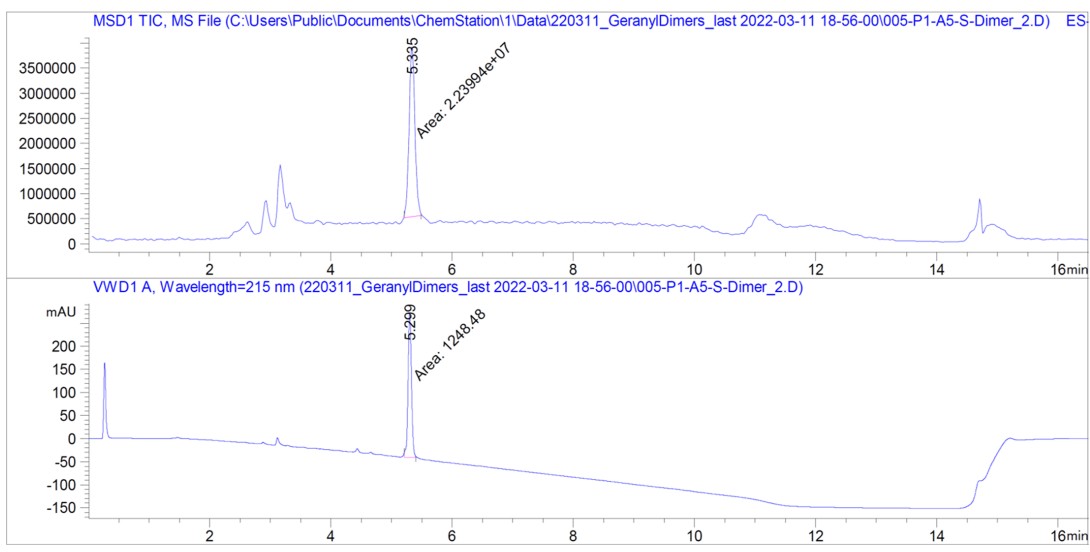

**Appendix 1—figure 5.** (Geranyl)$_2$-Lys2.

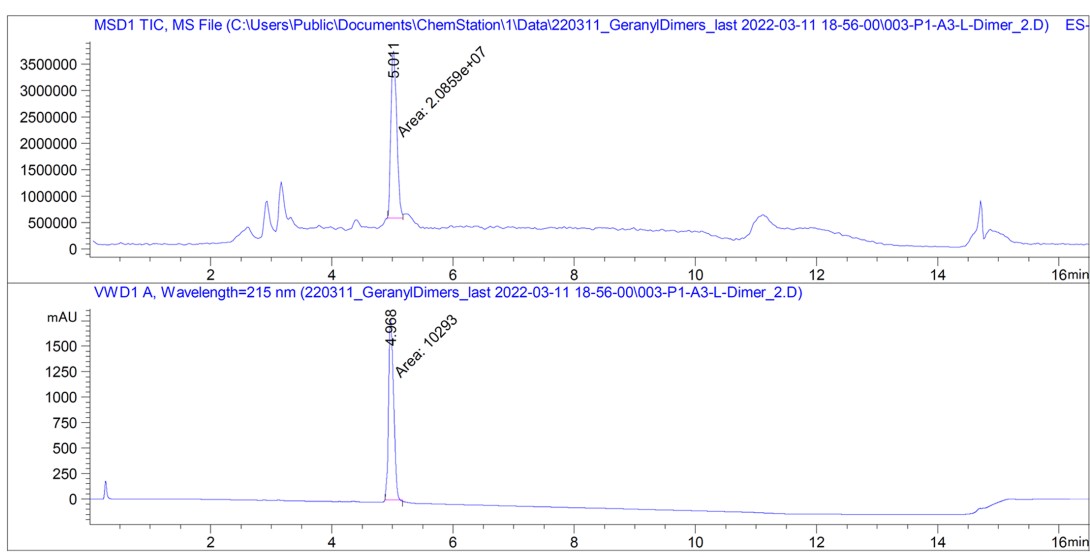

MSD1 TIC, MS File (C:\Users\Public\Documents\ChemStation\1\Data\220311_GeranylDimers_last 2022-03-11 18-56-00\003-P1-A3-L-Dimer_2.D)  ES-

5.011
Area: 2.0859e+07

VWD1 A, Wavelength=215 nm (220311_GeranylDimers_last 2022-03-11 18-56-00\003-P1-A3-L-Dimer_2.D)

4.968
Area: 10293

*MSD1 SPC, time=4.982:5.073 of C:\Users\Public\Documents\ChemStation\1\Data\220311_GeranylDimers_last 2022-03-11 18-56-00\003-P1-A3-L-I

Max: 226091

| Charge | $m_i$/z |
| --- | --- |
| -1 | 1315.75 |

1315.8
1316.8
1317.8
1318.8

MW 1317.58

**Appendix 1—figure 6.** (Geranyl-PEG3)$_2$-Lys.

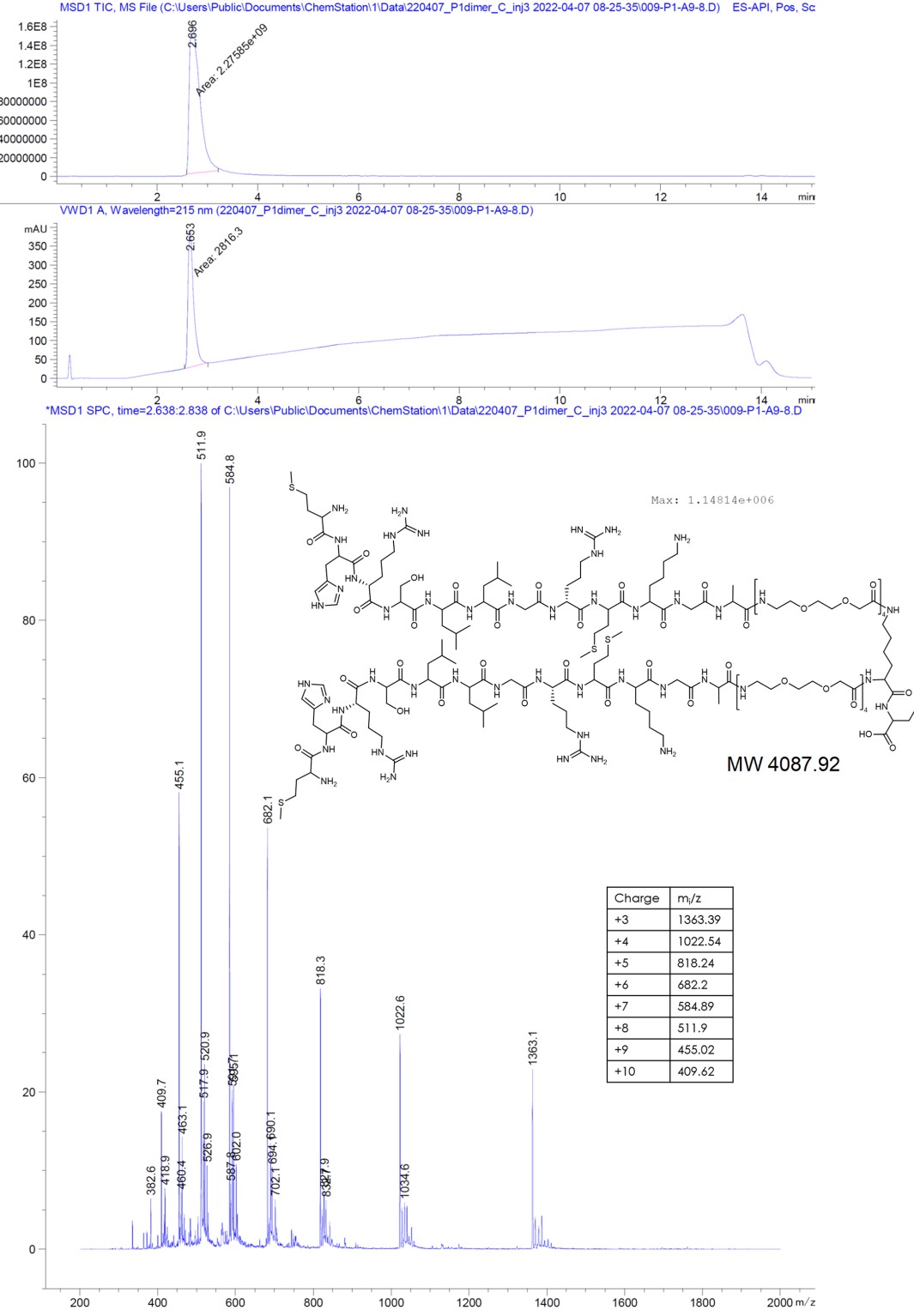

**Appendix 1—figure 7.** MHRSLLGRMKGA-PEGlinker-Dimer-C (P1dC).

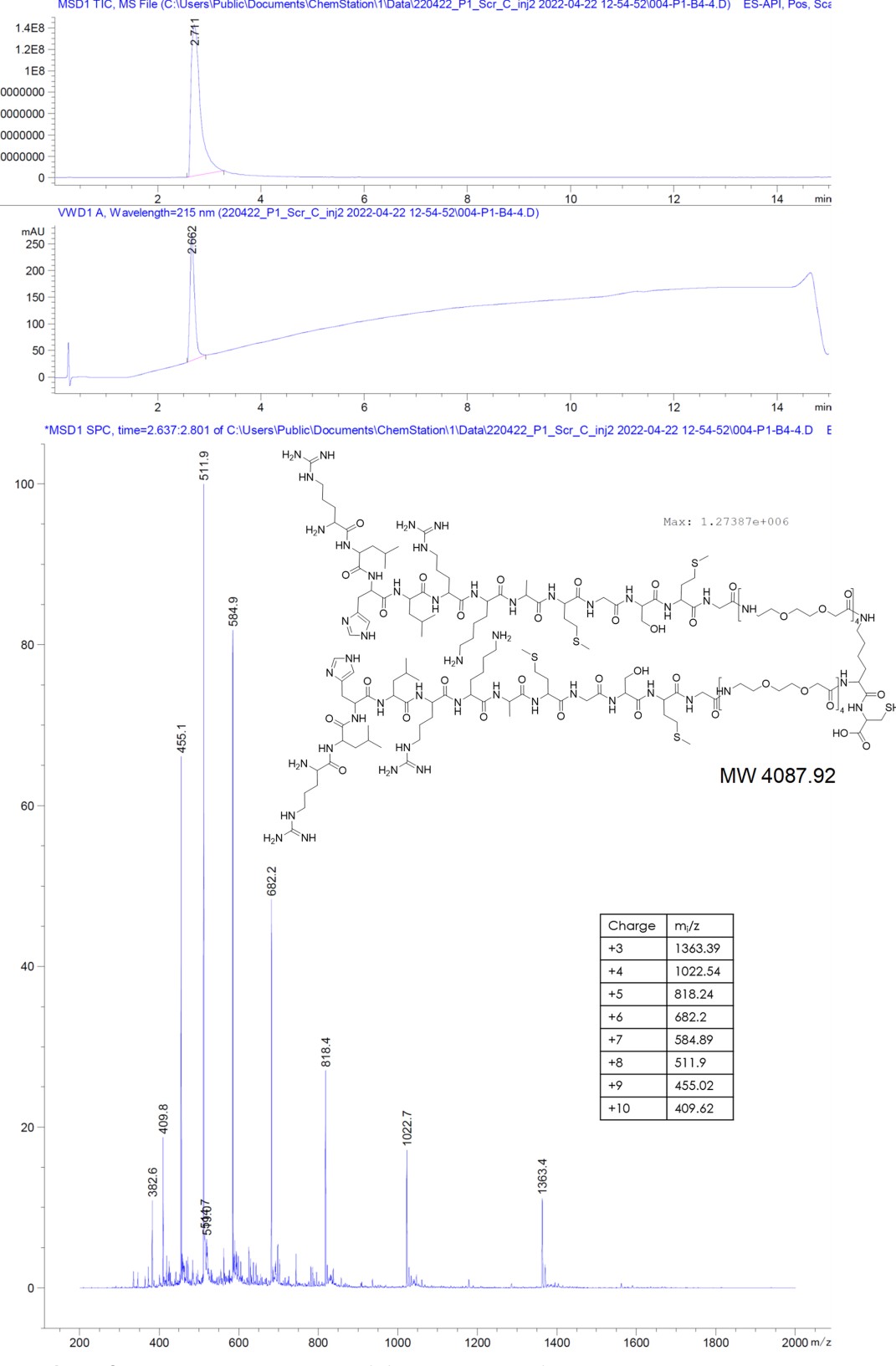

**Appendix 1—figure 8.** RLHLRKAMGSMG-PEGlinker-Dimer-C (scrP1dC).

MSD1 TIC, MS File (C:\Users\Public\Documents\ChemStation\1\Data\220401_TNB_CPP_inj5 2022-04-01 16-39-53\001-P1-C1-2.D)    ES-API, Pos, Sc

VWD1 A, Wavelength=215 nm (220401_TNB_CPP_inj5 2022-04-01 16-39-53\001-P1-C1-2.D)

*MSD1 SPC, time=0.238:0.310 of C:\Users\Public\Documents\ChemStation\1\Data\220401_TNB_CPP_inj5 2022-04-01 16-39-53\001-P1-C1-2.D

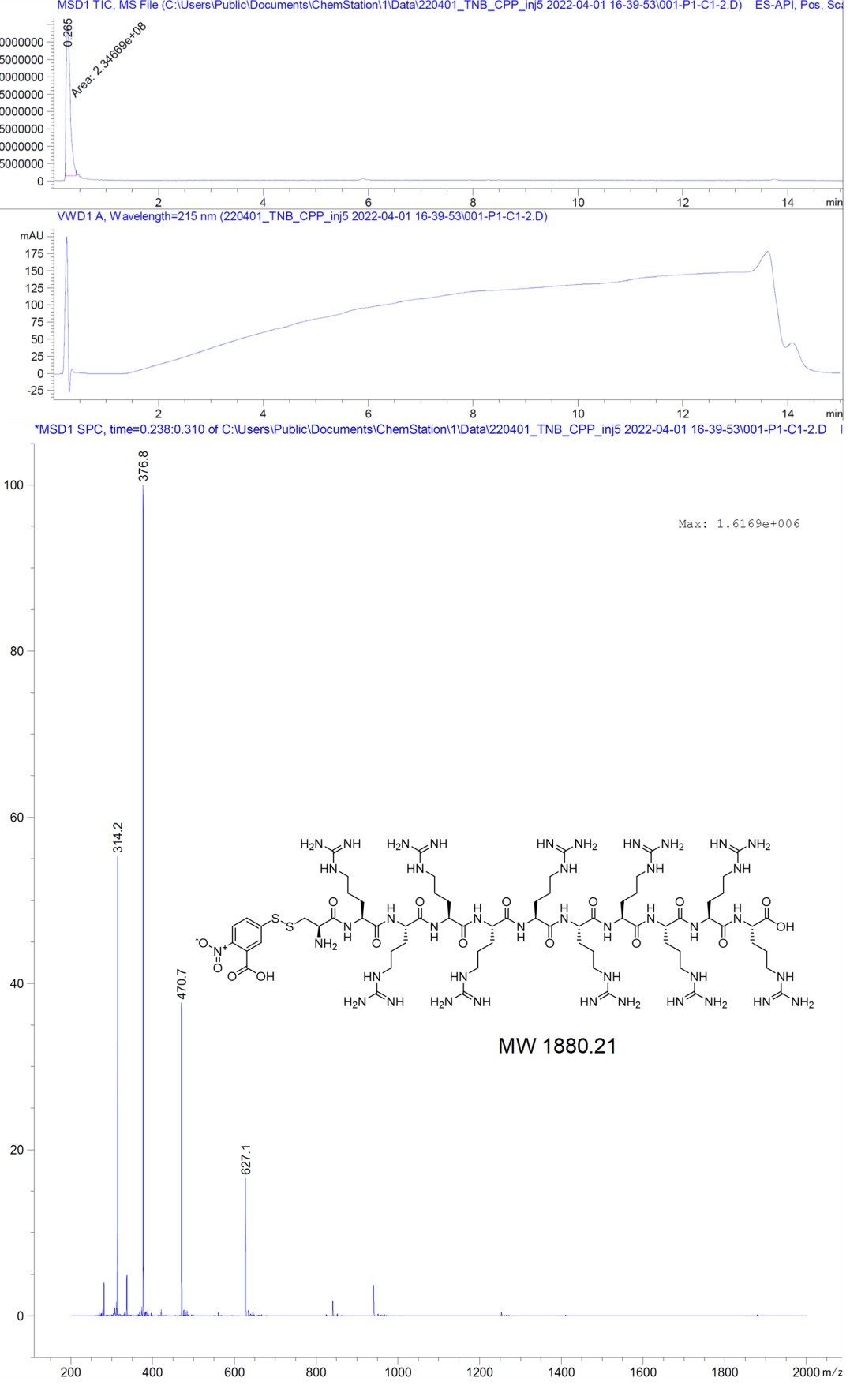

MW 1880.21

**Appendix 1—figure 9.** TNB-C-10R.

