## [Editor Report · eLife Assessment]

This **valuable** work presents an interesting strategy to interfere with the HBV infectious cycle as it identifies two previously unexplored HBc-Ag binding pockets. The experimental data is **compelling** and opens the door to generating and testing novel anti-HBV therapies.

---

## [Referee Report · Reviewer #1 (Public review)]

Summary:

In this paper, the authors present an interesting strategy to interfere with the HBV life cycle: the preparation of geranyl and peptides' dimers that could impede the correct assembly of hepatitis B core protein HBc into viable capsids. These dimers are of different nature, depending on the HBc site the authors plan to target. A preliminary study with geranyl dimers (targeting a hydrophobic site of HBc) was first investigated. The second series deals with peptide-PEG linker-peptide dimers, targeting the tips of HBc dimer spikes.

Strengths:

This work is very well conducted, combining ITC experiments (for determination of dimers' KD), cellular effects (thanks to the grafting of previously developed dimers with polyarginine-based cell penetrating peptide) HBV infected HEK293 cells and Cryo-EM studies.

The findings of these research teams unambiguously demonstrated the interest of such dimeric structures in impeding the correct HBV life cycle and thus, could bring solutions in the control of its development. Ultimately, a new class of HBV Capside Assembly Modulators could arise from this study.

There is no doubt that this work could bring very interesting information for people working on VHB.

Comments on revisions:

Minor corrections have been made in this revised version of this work, according to the remarks of the reviewers.

---

## [Referee Report · Reviewer #2 (Public review)]

Summary:

Vladimir Khayenko et al. discovered two novel binding pockets on HBc with in vitro binding and electron microscopy experiments. While the geranyl dimer targeting a central hydrophobic pocket displayed a micromolar affinity, the P1-dimer binding to the spike tip of HBc has a nanomolar affinity. In the turbidity assay and at the cellular level, an HBc aggregation from peptide crosslinking was demonstrated.

Strengths:

The study identifies two previously unexplored binding pockets on HBc capsids and develops novel binders targeting these sites with promising affinities.

Weaknesses:

While the in vitro and cellular HBc aggregation effects are demonstrated, the antiviral potential against HBV infection is not directly evaluated in this study.

---

## [Author Response]

The following is the authors’ response to the original reviews.

We appreciate the positive assessment and agree that the experimental data offer valuable insights into HBV capsid assembly inhibition. Based on the reviewers' suggestions, we have clarified the cryo-EM data and added structural and mechanistic details throughout the manuscript, which we believe significantly enhance its overall clarity and impact. The manuscript now better reflects a promising strategy to interfere with the HBV life cycle. We have carefully addressed all comments to improve both the clarity and quality of the manuscript.

**Response to Public Reviews**

We greatly appreciate the insightful comments and suggestions from the reviewers. Below, we provide responses to the points raised in the public reviews.

**Reviewer #1 (Public Review):**
Summary:In this paper, the authors present an interesting strategy to interfere with the HBV life cycle: the preparation of geranyl and peptides' dimers that could impede the correct assembly of hepatitis B core protein HBc into viable capsids. These dimers are of different nature, depending on the HBc site the authors plan to target. A preliminary study with geranyl dimers (targeting a hydrophobic site of HBc) was first investigated. The second series deals with peptide-PEG linker-peptide dimers, targeting the tips of HBc dimer spikes.Strengths:This work is very well conducted, combining ITC experiments (for determination of dimers' KD), cellular effects (thanks to the grafting of previously developed dimers with polyarginine-based cell penetrating peptide) HBV infected HEK293 cells and Cryo-EM studies.The findings of these research teams unambiguously demonstrated the interest of such dimeric structures in impeding the correct HBV life cycle and thus, could bring solutions in the control of its development. Ultimately, a new class of HBV Capside Assembly Modulators could arise from this study.There is no doubt that this work could bring very interesting information for people working on VHB.Weaknesses:Some minor corrections must be made, especially for a more precise description of the strategy and the chemical structure of the designed new VHB capsid assembly modulators.

We are grateful for the positive feedback on the experimental design, the combination of ITC, cellular effects, and Cryo-EM studies, and the potential for developing new classes of HBV Capsid Assembly Modulators (CAMs). In the revised version we have clarified the design rationale for the choice of the PEG linker length in the Supplementary Information, linking it to the structural measurements of the capsid. Chemical structures and detailed molecular formulas were added and terms have been corrected. A scrambled dimeric peptide served as a negative control, which showed no binding, confirming the specificity of our designed peptide and ruling out non-specific interactions from other elements of the molecules such as the linkers. Finally, we have revised the nomenclature for the geranyl dimers to better reflect the chemical structure. All figures, including Figure 3, have been updated to high-resolution. All mentioned typos have been corrected. Consultation dates have been added to the website references. HPLC terminology was corrected.

**Reviewer #2 (Public Review):**
Summary:Vladimir Khayenko et al. discovered two novel binding pockets on HBc with in vitro binding and electron microscopy experiments. While the geranyl dimer targeting a central hydrophobic pocket displayed a micromolar affinity, the P1-dimer binding to the spike tip of HBc has a nanomolar affinity. In the turbidity assay and at the cellular level, an HBc aggregation from peptide crosslinking was demonstrated.Strengths:The study identifies two previously unexplored binding pockets on HBc capsids and develops novel binders targeting these sites with promising affinities.Weaknesses:While the in vitro and cellular HBc aggregation effects are demonstrated, the antiviral potential against HBV infection is not directly evaluated in this study.

Thank you for recognizing the innovative approach of our work and the potential for developing novel antivirals targeting HBc. We have now included additional discussion on potential future experiments aimed at evaluating the compounds' effects on cellular physiology and viral infectivity.

**Reviewer #3 (public Review):**
Summary:HBV is a continuing public health problem and new therapeutics would be of great value. Khayenko et al examine two sites in the HBc dimer as possible targets for new therapeutics. Older drugs that target HBc bind at a pocket between two HBc dimers. In this study Khayenko et al examine sites located in the four helix bundle at the dimer interface.The first site is a pocket first identified as a triton100 binding site. The authors suggest it might bind terpenes and use geraniol as an example. They also test a decyl maltose detergent and a geraniol dimer intended for bivalent binding. The KDs were all in the 100µM range. Cryo-EM shows that geraniol binds the targeted site.The second site is at the tip of the spike. Peptides based on a 1995 study (reference 43) were investigated. The authors test a core peptide, two longer peptides, and a dimer of the longest peptide. A deep scan of the longest monomer sequence shows the importance of a core amino acid sequence. The dimeric peptide (P1-dimer) binds almost 100 fold better than the monomer parent (P1). Cryo-EM structures confirm the binding site. The dimeric peptide caused HBc capsid aggregation When HBc expressing cells were treated with active peptide attached to a cell penetrating peptide, the peptide caused aggregation of HBc antigen mirroring experiments with purified proteins.Strengths:The two sites have not been well investigated. This paper marks a start. The small collection of substrates investigated led to discovery of a dimeric peptide that leads to capsid aggregation, presumably by non-covalent crosslinking. The structures determined could be very useful for future investigations.Weaknesses:In this draft, the rational for targets for the triton x100 site is not well laid out. The target molecules bind with KDs weaker that 50µM. The way the structural results are displayed, one cannot be sure of the important features of binding site with respect to the the substrate. The peptide site and substrates are better developed, but structural and mechanistic details need to be described in greater detail.

We appreciate the reviewer’s positive comments on identifying and targeting previously unexplored sites on HBc, and the potential utility of our dimeric peptides in future studies. We have revised the Results section to better explain the rationale behind targeting the hydrophobic binding site. Additionally, the structures have been revised for clearer presentation, and we now emphasize the key features of the binding site and the role of substrate specificity.

**Recommendations For The Authors:**

**Reviewer #1 (Recommendations For The Authors):**
For clarity, the chemical structure of SLLGRM peptide, geraniol and HAP molecules must be indicated, preferably in Fig. 1 (at least in the Supplementary Information section).

We have now included the chemical structures of the SLLGRM peptide, geraniol, and HAP molecules for clarity in Figure 1 and in the main manuscript to ensure they are easily accessible for reference and to provide further detail and context.

In the same idea, in Fig. 1 (and in the text): The molecular formula of heteroaryldihydropyrimidine HAP must be clearly indicated, as the nature of the heteroatom (S, O, N?) in this "heteroaryl" derivative is not indicated.

The full molecular formula of HAP ((2S)-1-[[(4R)-4-(2-chloranyl-4-fluoranyl-phenyl)-5-methoxycarbonyl-2-(1,3-thiazol-2-yl)-1,4-dihydropyrimidin-6-yl]methyl]-4,4-bis(fluoranyl)-pyrrolidine-2-carboxylic acid), is now included the figure legend.

with a polyethylene glycol (PEG) linker that could bridge the distance of 38 Å between the two opposing hydrophobic pockets": what is the rationale of the design of this linker? Authors must explain briefly why/how they have chosen this linker length and nature (please indicate a reference for the appropriate choice of PEG linker). Same remarks for dimers targeting the capsid spike tips, having 50 angstroms PEG linkers. So, the choice of the linker length must be clearly explained and not be only mentioned in the sentence of the discussion part "Using our structural knowledge of the capsid, particularly the distances between the spikes.

We have now better clarified the rationale for the design of the PEG linker length. The linker lengths were specifically chosen based on structural knowledge of the capsid, particularly the measured distances between the spike tips (60 Å) and the hydrophobic pockets (40 Å). In the Supplementary Information (Supplementary Figure 1), we now clearly explain how these measurements guided the choice of PEG linker length, allowing for optimal bridging and interaction between the binding sites. This supplementary figure now explicitly connects the design rationale to the specific structural features of the capsid.

I do not agree with the authors when they claim a "nanomolar affinity of 312 nM". To me, a nanomolar affinity would require several of few tens of nanoM (but not three hundreds) ... So, please correct with "sub-micromolar affinity of 312 nM" and all the other parts of the manuscript (title and caption of Figure 3..., "the peptide dimer (P1dC) with nanomolar affinity" "nanomolar levels"...).

We thank the Rev#1 for pointing this out. Since the term "nanomolar affinity" can indeed be interpreted as referring to the lower end of the nanomolar range, rather than values close to 300 nM we have revised the manuscript to refer to the "sub-micromolar affinity" where applicable. This change has been made throughout the manuscript, including the subtitles and figure captions, and the text.

The drug design strategy was to combine two peptides showing low affinity, attached by a PEG linker with an appropriate length and appears obvious to me. But a control experiment is anyway missing: the peptide-PEG linker derivative (not the dimer peptide-PEG linker-peptide...) should have been evaluated for an unambiguous proof of concept of these dimeric peptides. To my opinion, for the publication of this work, these experiments should be brought (eg, when describing the affinities of SLLGR dimers). I agree that Cryo-EM experiments bring evidences of the dimer binding but the affinity values for (peptide-PEG linker) derivatives would bring an additional proof (as the PEG flexible linkers was not resolved by Cryo-EM).

Thank you for your thoughtful comment regarding the use of a monovalent control for the peptide-PEG linker. A scrambled dimeric peptide serves as a negative control. In ITC it showed no binding at all. Thereby ruling out possibly unspecific interactions mediated by the introduced PEG linker or handle itself.

Given the complete lack of binding with the scrambled dimeric peptide, we believe this thoroughly excludes the need for an additional monovalent control, as it provides strong evidence that the observed binding is driven specifically by the designed peptide sequence and not by the linker or other structural components. We have now made this clarification more explicit in the revised manuscript to avoid any ambiguity. We hope this addresses your concern, and we appreciate your suggestion to further strengthen the rigor of the work. Despite its identical charge, molecular weight and atom composition the scrambled control did not cause HBc aggregation in living cells, thus indicating sequence specific action of the aggregating dimer.

The nomenclature of the dimers must be modified because there is no logic between the name "long dimer" and the chemical structure. Particularly, the number of ethylene glycol motifs must be indicated: authors have to find an appropriate nomenclature indicating both the linker length and nature (small molecule or peptide) of the bivalent parts (and hence, do not mention anymore "short geranyl dimer" "long geranyl dimer").

Thank you for your valuable suggestion regarding the nomenclature of the dimers. We agree that the terms "short geranyl dimer" and "long geranyl dimer" do not fully reflect the chemical structure of the molecules. In response, we have revised the nomenclature to provide a clearer indication of both the linker length and the nature of the bivalent parts. We now refer to the dimers as (Geranyl)_2_-Lys for the dimer with two geranyl groups attached to lysine and (Geranyl-PEG3)_2_-Lys for the dimer with a PEG3 linker (three ethylene glycol units) between the lysine amine and the geranyl groups. These revised names more accurately describe the structural differences and should avoid any ambiguity.

Lines 198-199: "Among these, the dimerized P1 exhibited a higher 198 occupation of the binding site, as illustrated in Supplementary Figure 9." But in Supp. Fig. 9, dimer P1dC (10) is described. As the text above is describing P1-dimer (9), the Supp. Fig. 9 must be provided, if available. If not, please modify this conclusion accordingly. In the text, when mentioning dimerized P1 peptide, authors must indicate with which compound it deals: (9) or (10)?

Thank you for your careful reading of the manuscript and for pointing out the discrepancy. In Supplementary Figure 9, the dimer described is P1dC, not P1d. The text has been revised to clarify this. We appreciate your attention to detail.

Please note that the graphic quality of Figure 3 is bad as it results in pixelized drawings (especially for the chemical structures).

Thank you for your feedback regarding the quality of Figure 3. We have now updated all figures, including Figure 3, to high-resolution PNG format with 300-500 dpi to ensure optimal graphic quality. This should resolve the pixelization issue, particularly for the chemical structures.

Minor typos: "clinical studies, a third are CAMs.[6]" "to the spike base hydrophobic pocket" "geraniol affinity to the central hydrophobic pocket, we designed"

We have corrected the punctuation in the mentioned sentences and appreciate your careful review of the manuscript.

Concerning the citation of a website (references 5 and 6), I guess that the consultation date should be mentioned.

We have now updated the references accordingly, including the consultation dates.

In the Materials and Methods part, Peptide synthesis paragraph, authors must write "semi-preparative HPLC.

It’s now corrected to "semi-preparative HPLC".

In the supplementary information file, 1H and 13C NMR spectrum for the small molecule "Short Geranyl Dimer (SGD)" should be provided.

The purity and identity of this Geranyl derivate were confirmed through UV detection in LC-MS and supported by the mass spectra, which provide robust and clear evidence of the compound's structure and well-accepted method for confirming the structure in this context. While we understand the value of NMR in structural analysis, we believe that additional analytical evidence is not critical for this study.

**Reviewer #2 (Recommendations For The Authors):**
Overall, this study presents an innovative approach to target the HBV core protein and paves the way for developing new classes of antivirals with a distinct mechanism of action. The findings expand the current knowledge of druggable sites on HBc capsids and provide promising lead compounds. Future studies exploring the antiviral effects and optimizing the binders for therapeutic applications would be valuable next steps.

We sincerely thank the reviewer for the positive assessment of our work and for highlighting its innovative approach to targeting the HBV core protein. We appreciate your recognition of the study's potential in paving the way for developing new classes of antivirals with distinct mechanisms of action. Below, we provide responses to each of the points raised.

The significance of the central hydrophobic pocket as a target may require additional experiments for validation. Currently, the substrate binding activity is relatively low and appears to have a non-significant impact on HBc.

We agree that the central hydrophobic pocket exhibits relatively weak binding affinity with the ligands tested in this study. However, we have provided additional structural evidence and affinity data to support its relevance as a druggable site. In recognition of the weak affinity of these small molecules, we expanded our focus to include peptide-based binders, which yielded higher affinities, particularly when dimerized.

It might be more effective to present Figure 1B after summarizing all the results.

We understand the reviewer’s suggestion. However, we decided to highlight and summarize the major findings early in the manuscript. We included Figure 1B at the beginning to allow readers to quickly grasp the core concepts and outcomes of our study.

The labels for P1/P2 are presented in Figure 1A, yet their definitions are not provided until the second part of the Results section.

We appreciate the reviewer’s observation. While see a benefit of showing three trackable sites on HBV early and as an overview but we also agree that the early presentation of P1/P2 could lead to some confusion. To resolve this, we have revised the figure to introduce only on the minimal peptide to avoid any ambiguity. The full dimer sequences and names are introduced later.

Further investigation of the cytotoxic potential of peptide-induced HBc aggregation is necessary.

Investigating the cytotoxicity together with infectivity is an important future direction but outside the scope of this study. We now elaborate on this point in the discussion.

**Reviewer #3 (Recommendations For The Authors):**
Two sites in the dimer interface are shown to bind ligands. It is not shown that filling these regions will change infection. The exhaustive studies by Bruss showed point mutations directly alter infection and would be of value to discuss.

We thank Rev#3 for this very helpful comment. We now highlight how point mutations in these regions were shown to affect HBV infectivity. Thereby providing a link between our findings and how ligand binding might influence the viral life cycle.

It is not shown whether the two sites interact. Molecular dynamics by Hadden or Gumbart may be informative. The failure to look for a connection between these sites is an oversight.

We thank Rev#3 for the insightful suggestion to explore potential interactions between the two binding sites. We acknowledge that molecular dynamics (MD) simulations, such as those performed by Gumbart et al. and Hadden et al., could indeed provide valuable insights into the structural dynamics and potential cooperativity between these sites. Indeed, molecular dynamics of the HBV capsid by Perilla and Hadden has demonstrated significant flexibility in the capsid spikes and their interactions with neighboring subunits suggesting that the dynamics of binding sites could influence ligand accessibility and potential crosstalk.

We believe that our own previous structural studies together with data in this work provide substantial experimental evidence on this topic. In Makbul et al. 2021a (doi.org/10.3390/microorganisms9050956) we observed that peptide binding (particularly P2) did not stabilize the spikes; instead, the upper part of the spikes exhibited considerable wobbling. This variability mirrored the conformational diversity reported in MD simulations. Using local classification, we noted that the variability in the spike's upper region was greater when P2 was bound than in its absence. Additionally, in Makbul et al. 2021b (doi.org/10.3390/v13112115), we showed that peptide binding had little effect on the hydrophobic pocket beneath the mobile spike region, located in the more rigid part of the capsid. While we observed F97 in the D-monomer adopting two alternate rotamer orientations upon P2 binding this was not exclusive to P2, as similar changes were noted in the L60V mutant even without bound peptide.

We have updated the manuscript to briefly discuss this crosstalk, that provides additional context to our findings. Interestingly, only TX100—but not geraniol—completely flipped F97 into an alternate orientation, forming a new π-π stacking interaction with the mobile region of the spike. This finding suggests that interactions within the hydrophobic pocket are transmitted based on ligand specific interactions to the tips of the spikes. Thus, supporting and refining the concept of a crosstalk between binding sites, primarily initiated from the hydrophobic pocket in a ligand specific fashion.

The logic for proposing a terpene ligand is strained. Comparisons are made to HBs and the HDV delta antigen. However, HBs is myristoylated not farnesylated and delta antigen binds HBs not HBc.

We have revised the text to clarify the rationale for testing terpenes as ligands, focusing instead on the specific properties of the hydrophobic pocket targeted by geraniol.

The authors suggest larger terpenes as binding agents, but there does not appear to be room for a longer molecule in the binding site. The authors do not discuss whether a longer molecule could be modeled in the site based on their density.

We appreciate this observation and agree that the potential for larger terpenes to bind this site is not obvious from the structural data presented in this work. We have now included a more detailed visualization (Fig2D) and discussion of the hydrophobic binding pocket, based on the density observed in the presented geraniol structure and the previous triton structure and discuss its implications of the binding of larger hydrophobic molecules into the site (Fig 2D).

The authors note that the structure could explain molecular details of this site, but these are not discussed. A more complete analysis of the geraniol protein is necessary, including an estimate of the resolution of that density.

We agree that a more complete analysis of the hydrophobic binding site was warranted. We have now expanded the discussion of the structural details of this binding site based on the geraniol-bound structure, the density and occupancy accounted by this ligand. These additional details (Fig 2C,D and Fig 5) should provide a clearer understanding of the binding interactions observed.

The dimeric geraniol is marginally better binding than the monomer, two-fold, but this could be due to doubling the number of geraniols per ligand or due to an undefined interaction of the extended molecule with the surface of the capsid. A geraniol linker should be tested.

The modest improvement in binding may indeed only reflect the doubled number of geraniols rather than linker-mediated avidity effects. Interaction of the linker with the capsid surface is ruled-out by the scrambled control that included the same linkers but did not show any capacity to bind.

Is the enhanced binding of dimer due to bivalent binding of dimer to one capsid? Is it a chance interaction of the linker with the surface of HBc, which is easily tested? Is it an avidity effect due to aggregation of capsids?

Thank you for this insightful question. Our data suggest that the enhanced binding is due to bivalent interactions. To address the possibility of non-specific interactions from either the handle or the linker, we included a scrambled dimeric peptide as a negative control, which showed no binding. This rules out non-specific interactions from the linker or handle. Given this, we believe an additional monovalent control is unnecessary, as the scrambled control confirms that the binding is driven by the geraniol and peptide warheads alone. We have clarified this in the revised manuscript and appreciate your suggestion to strengthen the study.

The experimental analysis of point mutation of P1 is not analyzed beyond stating that it shows the importance of the core peptide sequence. Is there rationale for the effect of R3 to E and K10 to E mutation?

We appreciate the reviewer's curiosity and request for a more detailed discussion of the P1 deep mutational scan data and its implications. The observed low mutation tolerance of the core peptide sequence SLLGRM regarding HBc binding is highly consistent with our prior structural data and binding studies in solutions (https://doi.org/10.3390/microorganisms9050956) as well as the results from the original phage library screening (M. R. Dyson, K. Murray, Proceedings of the National Academy of Sciences 1995, 92, 2194–2198), and the binding data presented here. Notably, the data set does not suggest that additional binding interfaces contribute to the aggregation seen with N-terminal elongated P1 and P2 versus the non-aggregating shorter SLLGRM. While the positional scan largely aligns with previous phage binding hierarchy and quantified ligands, we were previously prompted by surprising affinity gains for positive to negative amino exchanges in related peptides in same way as Rev#3: Specifically, “SLLGEM” has been predicted previously and here to show enhanced affinity over “SLLGRM”. Quantification in solution, however, could not confirm this enhanced HBV binding affinity (Makbul et al. 2021 Microorganisms), which could not be recapitulated by in solution quantification. In the revised version of the manuscript we now highlight the possible limited predictive power of this assay for positions where positively charged residues are exchanged by negatively charged residues (Figure legend of Fig 3D).

The fluctuations in Figure 3B could be largely magnification of noise due to changing the y-axis. The fluctuations can be characterized as standard variation, excluding the injections, to allow a quantitative judgment.

Isothermal titration calorimetry heat fluctuations without injections are now shown in the supplementary information scaled to the same y-axis (Supplementary Figure 3D).

Molecular graphics throughout are too small and poorly labeled.

We have revised the molecular graphics throughout the manuscript to increase their size and improve labeling for clarity. All figures are now provided in 500dpi.

In Figure 2, compounds 1 and 2 are pyrophosphates. The label in the figure should be corrected.

Thank you for pointing this out. These compounds were removed for clarity.

In the introduction, the phrase "discontinuation frequently leads to relapse" should be changed to something less ambiguous.

Thank you for highlighting this point regarding the phrasing in the introduction. We have revised the statement to more accurately reflect the clinical situation by specifying that stopping treatment often results in viral rebound and disease recurrence in many patients. This adjustment clarifies the intended meaning and addresses the ambiguity you identified. We hope this revision better aligns with the clinical context of HBV management and improves the overall clarity of the manuscript.

Define "functional cure" in the introduction.

Thank you for your suggestion to clarify the term 'functional cure.' We have revised the manuscript and instead of ”functional cure” we mention the goal of sustained viral suppression without detectable HBV DNA and loss of hepatitis B surface antigen (HBsAg) without the need for continuous therapy. This should provide greater clarity for readers and improve the overall comprehensibility of the introduction.

The sentence beginning line 92 is not clear unless one has already read the paper. Figure 1 is not well described.

Thank you for your valuable feedback regarding the clarity of this sentence and the legend of Figure 1. We have revised the text and legend to provide more context and improve the flow for readers who are unfamiliar with the specifics of the study. The revised version now clearly explains the targeted binding sites and the purpose of the bivalent binders at the beginning of the results section.

In line 235 the meaning is not clear. What is in excess? Is there free CPP in solution? Is it the charge on the CPP?

We have clarified the passage as requested.

When describing peptide-induced aggregation, Figures 5 and 6, figure 1B is never referred to. Figure 1B would work better as part of Figure 6.

We understand the reviewer’s suggestion. However, we decided to highlight and summarize the major findings and the underlying hypothesis early in the manuscript. We included Figure 1B at the beginning to allow readers to quickly grasp a core concept and outcome of our study.

We now however refer to Figure 1B and together with all the other changes hope that we have improved the clarity and quality of the manuscript.

We appreciate your constructive feedback and the opportunity to further refine the work.